# On Large-Cohort Training for Federated Learning

**Zachary Charles**
Google
zachcharles@google.com

**Zachary Garrett**
Google
zachgarrett@google.com

**Zhouyuan Huo**
Google
zhhuo@google.com

**Sergei Shmulyian**
Google
sshmulyian@google.com

**Virginia Smith**
Carnegie Mellon University
smithv@cmu.edu

## Abstract

Federated learning methods typically learn a model by iteratively sampling updates from a population of clients. In this work, we explore how the number of clients sampled at each round (the *cohort size*) impacts the quality of the learned model and the training dynamics of federated learning algorithms. Our work poses three fundamental questions. First, what challenges arise when trying to scale federated learning to larger cohorts? Second, what parallels exist between cohort sizes in federated learning, and batch sizes in centralized learning? Last, how can we design federated learning methods that effectively utilize larger cohort sizes? We give partial answers to these questions based on extensive empirical evaluation. Our work highlights a number of challenges stemming from the use of larger cohorts. While some of these (such as generalization issues and diminishing returns) are analogs of large-batch training challenges, others (including catastrophic training failures and fairness concerns) are unique to federated learning.

## 1 Introduction

Federated learning (FL) [52] considers learning a model from multiple clients without directly sharing training data, often under the orchestration of a central server. In this work we focus on *cross-device* FL, in which the aim is to learn across a large population of edge devices [27, Table 1]. A distinguishing characteristic of cross-device FL is *partial participation* of the client population: Due to systems constraints such as network size, the server typically only communicates with a subset of the clients at a time[1]. For example, in the popular `FedAvg` algorithm [52], at each communication round the server broadcasts its current model to a subset of available clients (referred to as a *cohort*), who use the model to initialize local optimization and send their model updates back to the server.

Intuitively, larger cohort sizes have the potential to improve the convergence of FL algorithms. By sampling more clients per round, we can observe a more representative sample of the underlying population—possibly reducing the number of communication rounds needed to achieve a given accuracy. This intuition is reflected in many convergence analyses of FL methods [29, 31, 32, 62, 69], which generally show that asymptotic convergence rates improve as the cohort size increases.

Larger cohorts can also provide privacy benefits. For example, when using the distributed differential privacy model [6, 11, 16, 65] in federated learning, noise is typically added to the updates sent from the clients to the server [54]. This helps preserve privacy but can also mar the utility of the learned model. By dividing the noise among more clients, larger cohorts may mitigate detrimental effects of noise. Moreover, since privacy tends to decrease as a function of the number of communication

---

[1]In contrast, *cross-silo* settings often have a small set of clients, most of which participate in each round [27].

35th Conference on Neural Information Processing Systems (NeurIPS 2021).

rounds [1, 17], larger cohorts also have the potential to improve privacy in FL by reducing the number of rounds needed for convergence.

Motivated by the potential benefits of large-cohort training, we systematically explore the impact of cohort size in realistic cross-device settings. Our results show that increasing the cohort size may not lead to significant convergence improvements in practice, despite their theoretical benefit [69]. Moreover, large-cohort training can introduce fundamental optimization and generalization issues. Our results are reminiscent of work on large-batch training in centralized settings, where larger batches can stagnate convergence improvements [14, 19, 51, 70, 71], and even lead to generalization issues with deep neural networks [23, 30, 46–48, 50, 64]. While some of the challenges we identify with large-cohort training are parallel to issues that arise in large-batch centralized learning, others are unique to federated learning and have not been previously identified in the literature.

**Contributions.** In this work, we provide a novel examination of cohort sizes in federated learning. We give a wide ranging empirical analysis spanning many popular federated algorithms and datasets (Section 2). Despite the many possible benefits of large-cohort training, we find that challenges exist in realizing these benefits (Section 3). We show that these issues are caused in part by distinctive characteristics of federated training dynamics (Section 4). Using these insights, we provide partial solutions to the challenges we identify (Section 5), focusing on how to adapt techniques from large-batch training, and the limitations of such approaches. Our solutions are designed to serve as simple benchmarks for future work. We conclude by discussing limitations and open problems (Section 6). Throughout, we attempt to uncover interesting theoretical questions, but remain firmly grounded in the practical realities of federated learning.

## 1.1   Related Work

**Large-batch training.** In non-federated settings, mini-batch stochastic gradient descent (SGD) and its variants are common choices for training machine learning models, particularly deep neural networks. While larger mini-batch sizes ostensibly allow for improved convergence (in terms of the number of steps required to reach a desired accuracy), in practice speedups may quickly saturate when increasing the mini-batch size. This property of diminishing returns has been explored both empirically [14, 19, 51, 64] and theoretically [48, 70]. Beyond the issue of speedup saturation, numerous works have also observed a *generalization gap* when training deep neural networks with large batches [23, 30, 46, 47, 50, 71]. Our work differs from these areas by specifically exploring how the cohort size (the number of selected clients) affects *federated* optimization methods. While some of the issues with large-batch training appear in large-cohort training, we also identify a number of new challenges introduced by the federated setting.

**Optimization for federated learning.** Significant attention has been paid towards developing federated optimization techniques. Such work has focused on various aspects, including communication-efficiency [5, 34, 37, 52], data and systems heterogeneity [25, 28, 29, 39–42, 67], and fairness [26, 43]. We provide a description of some relevant methods in Section 2, and defer readers to recent surveys such as [27] and [41] for additional background. One area pertinent to our work is that of variance reduction for federated learning, which can mitigate negative effects of data heterogeneity [28, 29, 73]. However, such methods often require clients to maintain state across rounds [29, 73], which may be infeasible in cross-device settings [27]. Moreover, such methods may not perform well in settings with limited client participation [62]. Many convergence analyses of federated optimization methods show that larger cohort sizes can lead to improved convergence rates, even without explicit variance reduction [31, 32, 69]. These analyses typically focus on asymptotic convergence, and require assumptions on learning rates and heterogeneity that may not hold in practice [9, 27]. In this work, we attempt to see whether increasing the cohort size leads to improved convergence in practical, communication-limited settings.

**Client sampling.** A number of works have explored how to select cohorts of a fixed size in cross-device FL [10, 12, 18, 59, 63]. Such methods can yield faster convergence than random sampling by carefully selecting the clients that participate at each round, based on quantities such as the client loss. However, such approaches typically require the server to be able to choose which clients participate in a cohort. In practice, cohort selection in cross-device federated learning is often governed by client availability, and is not controlled by the server [7, 61]. In this work we instead focus on the impact of size of the cohort, assuming the cohort is sampled at random.

## 2 Preliminaries

Federated optimization methods often aim to minimize a weighted average of client loss functions:

$$\min_x f(x) := \sum_{k=1}^{K} p_k f_k(x), \tag{1}$$

where $K$ is total number of clients, the $p_k$ are client weights satisfying $p_k \geq 0$, and $f_k$ is the loss function of client $k$. For practical reasons, $p_k$ is often set to the number of examples in client $k$'s local dataset [42, 52].

To solve (1), each client in a sampled cohort could send $\nabla f_k(x)$ to the server, and the server could then apply (mini-batch) SGD. This approach is referred to as FedSGD [52]. This requires communication for every model update, which may not be desirable in communication-limited settings. To address this, McMahan et al. [52] propose FedAvg, in which clients perform multiple epochs of local training, potentially reducing the number of communication rounds needed for convergence.

We focus on a more general framework, FedOpt, introduced by Reddi et al. [62] that uses both client and server optimization. At each round, the server sends its model $x$ to a cohort of clients $C$ of size $M$. Each client $c_k \in C$ performs $E$ epochs of training using mini-batch SGD with client learning rate $\eta_c$, producing a local model $x_k$. Each client $k \in C$ then communicates their *client update* $\Delta_k$ to the server, where $\Delta_k := x_k - x$ is the difference between the client's local model and the server model. The server computes a weighted average $\Delta$ of the client updates, and updates its own model via

$$x' = \text{SERVEROPT}(x, \eta_s, \Delta), \tag{2}$$

where $\text{SERVEROPT}(x, \eta_s, g)$ is some first-order optimizer, $\eta_s$ is the server learning rate, and $g$ is a gradient estimate. For example, if SERVEROPT is SGD, then $\text{SERVEROPT}(x, \eta_s, g) = x - \eta_s g$. The $\Delta$ in (2) is referred to as a **pseudo-gradient** [62]. While $\Delta$ may not be an unbiased estimate of $\nabla f$, it can serve a somewhat comparable role (though as we show in Section 4, there are important distinctions). Full pseudo-code of FedOpt is given in Algorithm 1.

---

**Algorithm 1** FedOpt framework

---

**Input:** $M, T\ E, x^1, \eta_c, \eta_s, \text{SERVEROPT}, \{p_k\}_{k=1}^K$
**for** $t = 1, \cdots, T$ **do**
    The server selects a cohort $C_t$ of $M$ clients uniformly at random, without replacement.
    The server sends $x^t$ to all clients in $C_t$.
    Each client $k \in C_t$ performs $E$ epochs of mini-batch SGD on $f_k$ with step-size $\eta_c$.
    After training, each $k \in C_t$ has a local model $x_k^t$ and sends $\Delta_k^t = x^t - x_k^t$ to the server.
    The server computes a pseudo-gradient $\Delta^t$ and updates its model via

$$\Delta^t = \frac{\sum_{k \in C_t} p_k \Delta_k^t}{\sum_{k \in C_t} p_k}, \quad x^{t+1} = \text{SERVEROPT}(x_t, \eta_s, \Delta^t).$$

---

Algorithm 1 generalizes a number of federated learning algorithms, including FedAvg [52], FedAvgM [25], FedAdagrad [62], and FedAdam [62]. These are the cases where SERVEROPT is SGD, SGD with momentum, Adagrad [15, 53], and Adam [33], respectively. FedSGD is realized when SERVEROPT is SGD, $\eta_c = 1$, $E = 1$, and each client performs full-batch gradient descent.

### 2.1 Experimental Setup

We aim to understand how the cohort size $M$ impacts the performance of Algorithm 1. In order to study this, we perform a wide-ranging empirical evaluation using various special cases of Algorithm 1 across multiple datasets, models, and tasks. We discuss the key facets of our experiments below.

**Datasets, models, and tasks.** We use four datasets: CIFAR-100 [35], EMNIST [13], Shakespeare [8], and Stack Overflow [3]. For CIFAR-100, we use the client partitioning proposed by Reddi et al. [62]. The other three datasets have natural client partitions that we use. For EMNIST, the handwritten characters are partitioned by their author. For Shakespeare, speaking lines in Shakespeare plays are

Table 1: Dataset statistics.

| DATASET | TRAIN CLIENTS | TRAIN EXAMPLES | TEST CLIENTS | TEST EXAMPLES |
|---|---|---|---|---|
| CIFAR-100 | 500 | 50,000 | 100 | 10,000 |
| EMNIST | 3,400 | 671,585 | 3,400 | 77,483 |
| SHAKESPEARE | 715 | 16,068 | 715 | 2,356 |
| STACK OVERFLOW | 342,477 | 135,818,730 | 204,088 | 16,586,035 |

partitioned by their speaker. For Stack Overflow, posts on the forum are partitioned by their author. The number of clients and examples in the training and test sets are given in Table 1.

For CIFAR-100, we train a ResNet-18, replacing batch normalization layers with group normalization (as proposed and empirically validated in federated settings by Hsieh et al. [24]). For EMNIST, we train a convolutional network with two convolutional layers, max-pooling, dropout, and two dense layers. For Shakespeare, we train an RNN with two LSTM layers to perform next-character-prediction. For Stack Overflow, we perform next-word-prediction using an RNN with a single LSTM layer. For full details on the models and datasets, see Appendix A.1.

**Algorithms.** We implement many special cases of Algorithm 1, including FedSGD, FedAvg, FedAvgM, FedAdagrad, and FedAdam. We also develop two novel methods: FedLARS and FedLamb, which are the special cases of Algorithm 1 where SERVEROPT is LARS [71] and Lamb [72], respectively. See Section 5 for the motivation and full details of these algorithms.

**Implementation and tuning.** Unless otherwise specified, in Algorithm 1 clients perform $E = 1$ epochs of training with mini-batch SGD. Their batch size is fixed per-task. We set $p_k$ to be the number of examples in client $k$'s dataset. We tune learning rates for all algorithms and models using a held-out validation set: We perform $T = 1500$ rounds of training with $M = 50, E = 1$ for each algorithm and model, varying $\eta_c, \eta_s$ over $\{10^i \mid -3 \le i \le 1\}$ and select the values that maximize the average validation performance over 5 random trials. All other hyperparameters (such as momentum) are fixed. For more details, see Appendix A. We provide open-source implementations of all simulations in TensorFlow Federated [4][2]. All experiments were conducted using clusters of multi-core CPUs, though our results are independent of wall-clock time and amount of compute resources.

**Presentation of results.** We apply the algorithms above to the tasks listed above with varying cohort sizes. For brevity, we present only a fraction of our results, selecting representative experiments to illustrate large-cohort training phenomena. The full set of experimental results can be found in Appendix B. We run 5 random trials for each experiment, varying the model initialization and which clients are sampled per round. In all subsequent figures, dark lines indicate the mean across the 5 trials, and shaded regions indicate one standard deviation above and below the mean.

## 3 Large-Cohort Training Challenges

In this section we explore challenges that exist when using large cohorts in federated learning. While some of these challenges mirror issues in large-batch training, others are unique to federated settings. While we provide concrete recommendations for mitigating some of these challenges, our discussion is generally centered around introducing and exploring these challenges in the context of federated learning.

### 3.1 Catastrophic Training Failures

We first discuss a practical issue unique to large-cohort training. Due to data heterogeneity, the server model $x$ may be misaligned with some client's loss $f_k$, in which case $\nabla f_k(x)$ can blow up and lead to optimization problems. This issue is exacerbated by large cohorts, as we are more likely to sample misaligned clients. To demonstrate this, we applied FedAvg with varying cohort sizes $M$, using learning rates tuned for $M = 10$. For each $M$, we performed 5 random trials and recorded whether a *catastrophic training failure* occurred, in which the training accuracy decreased by a factor of at least $1/2$ in a single round.

---

[2]`https://github.com/google-research/federated/tree/f4e26c1b9b47ac320e520a8b9943ea2c5324b8c2/large_cohort`

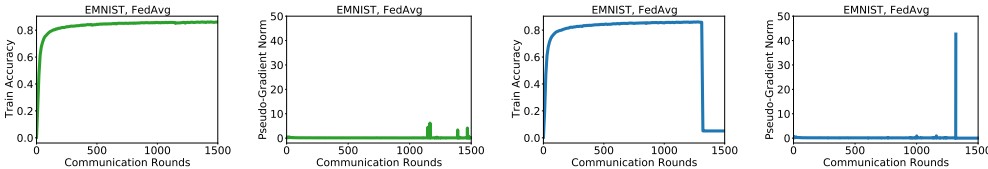

Figure 1: Applying `FedAvg` to EMNIST with cohort size 200. We plot the train accuracy and norm of the pseudo-gradient for a trial that ran successfully (**left**), and one that experienced a catastrophic training failure (**right**). The trials differed only in which clients were randomly sampled each round.

On EMNIST, the failure rate increased from 0% for $M = 10$ to 80% for $M = 800$. When failures occurred, we consistently saw a spike in the norm of the pseudo-gradient $\Delta$ (see Figure 1). These trends occurred on all datasets. In order to prevent this spike, we apply clipping to the client updates. We use the *adaptive clipping* method of [2]. While this technique was originally designed for training with differential privacy, we found that it greatly improved the stability of large-cohort training. Applying `FedAvg` to EMNIST with adaptive clipping, no catastrophic training failures occurred for any cohort size. We use adaptive clipping in all subsequent experiments. For more details, see Appendix A.3.

## 3.2 Diminishing Returns

In this section, we show that increasing $M$ in Algorithm 1 can lead to improved convergence, but that such improvements diminish with $M$. To demonstrate this, we plot the test accuracy of `FedAvg` and `FedSGD` across multiple tasks, for varying cohort sizes $M$. Results for CIFAR-100 and Stack Overflow are given in Figure 2, though we observe similar trends for all tasks (Appendix B.1).

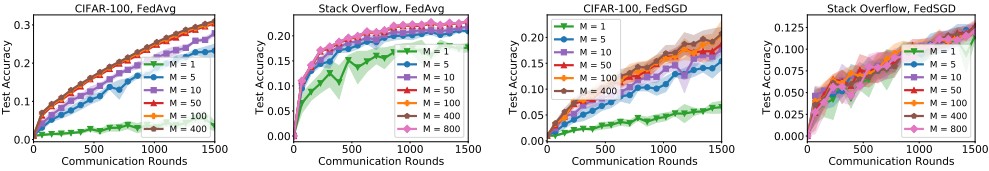

Figure 2: Test accuracy of `FedAvg` (top) and `FedSGD` (bottom), for various cohort sizes $M$, over the course of 1500 communication rounds.

We see that convergence benefits do not scale linearly with cohort size. While increasing $M$ from 1 to 10 can significantly improve convergence, there is generally a threshold after which point increasing $M$ incurs little to no change in convergence. This threshold is typically between $M = 10$ and $M = 50$. Interestingly, this seems to be true for both tasks, even though $M = 50$ represents 10% of the training clients for CIFAR-10, but only approximately 0.015% of the training clients for Stack Overflow. We see comparable results for EMNIST and Shakespeare, as well as for other optimizers, including `FedAdam` and `FedAdagrad`. See Appendix B.1 for the full results. In short, we see that increasing $M$ alone can lead to *diminishing returns*, or even no returns in terms of convergence. This mirrors issues of diminishing returns in large-batch training [14, 19, 51, 64].

## 3.3 Generalization Failures

Large-batch centralized optimization methods have repeatedly been shown to converge to models with worse generalization ability than models found by small-batch methods [23, 30, 46, 47, 50, 71]. Given the parallels between batch size in centralized learning and cohort size in FL, this raises obvious questions about whether similar issues occur in FL. In order to test this, we applied `FedAvg`, `FedAdam`, and `FedAdagrad` with different cohort sizes to various models. In Figure 3 we plot the train and test accuracy of our models after $T = 1500$ communication rounds of `FedAvg`, `FedAdam`, and `FedAdagrad`.

We find that generalization issues do occur in FL. For example, consider `FedAdam` on the CIFAR-100 task. While it attains roughly the same training accuracy for $M \in \{50, 100, 200, 400\}$, we see that

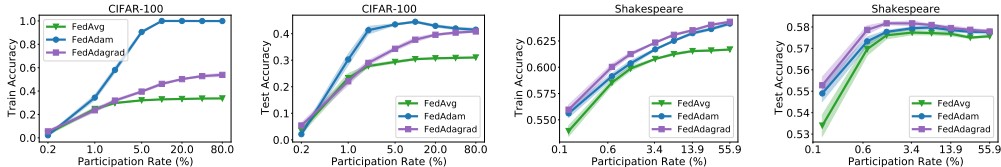

Figure 3: The train accuracy and test accuracy of `FedAvg`, `FedAdam`, and `FedAdagrad` on CIFAR-100 (left) and Shakespeare (right) after training for 1500 communication rounds, for varying cohort sizes. The $x$-axis denotes the percentage of training clients in each cohort.

the larger cohorts uniformly lead to worse generalization. This resembles the findings of Keskar et al. [30], who show that generalization issues of large-batch training can occur even though the methods reach similar training losses. However, generalization issues do not occur uniformly. It is often optimizer-dependent (as in CIFAR-100) and does not occur on the EMNIST and Stack Overflow datasets (see Appendix B.2). Notably, CIFAR-100 and Shakespeare have many fewer clients overall. Thus, large-cohort training may reduce generalization, especially when the cohort size is large compared to the total number of clients.

## 3.4 Fairness Concerns

One critical issue in FL is fairness across clients, as minimizing (1) may disadvantage some clients [43, 56]. Intuitively, large-cohort training methods may be better suited for ensuring fairness, since a greater fraction of the population is allowed to contribute to the model at each round. As a coarse measure of fairness, we compute percentiles of accuracy of our trained models across test clients. Under many notions of fairness, this would lead to higher accuracy values for smaller percentiles. The percentiles for `FedAdam` on each task are given in Figure 4.

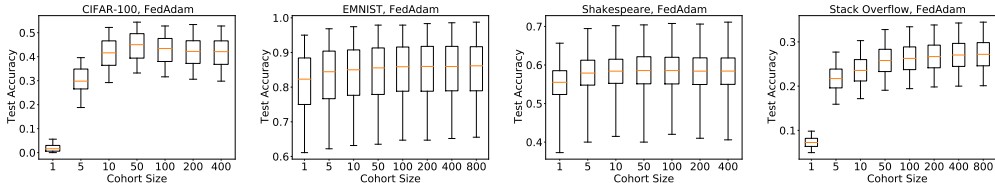

Figure 4: Accuracy of `FedAdam` after training for 1500 communication rounds using varying cohort sizes and tasks. The box plots show the 5th, 25th, 50th, 75th, and 95th percentiles of accuracy across test clients.

We find that the cohort size seems to affect all percentiles in the same manner. For example, on CIFAR-100, $M = 50$ performs better for smaller percentiles and larger percentiles than larger $M$. This mirrors the CIFAR-100 generalization failures from Section 3.3. By contrast, for Stack Overflow we see increases in all percentiles as we increase $M$. While the accuracy gains are only slight, they are consistent across percentiles. This suggests a connection between the fairness of a federated training algorithm and the fraction of test clients participating at every round. Notably, increasing $M$ seems to have little effect on the spread between percentiles (such as the difference between the 75th and 25th percentiles) beyond a certain point. See Appendix B.4 for more results.

## 3.5 Decreased Data Efficiency

Despite issues such as diminishing returns and generalization failures, federated optimization methods can see some benefit from larger cohorts. Large-cohort training, especially with adaptive optimizers, often leads to faster convergence to given accuracy thresholds. For example, in Figure 5, we see that the number of rounds `FedAdam` requires to reach certain accuracy thresholds generally decreases with the cohort size.

While it is tempting to say that large-cohort methods are "faster", this ignores the practical costs of large-cohort training. Completing a single communication round often requires more resources with larger cohorts. To showcase this, we also plot the accuracy of `FedAdam` with respect to the number of

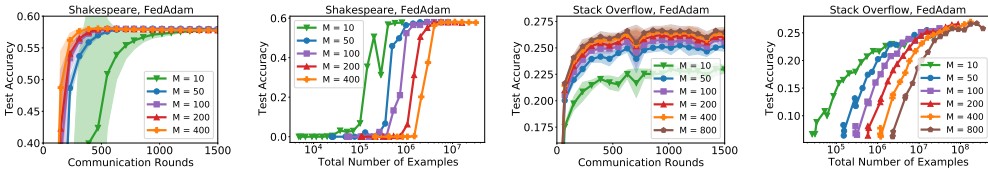

Figure 5: Test accuracy of `FedAdam` on Shakespeare (left) and Stack Overflow (right) with various cohort sizes. We plot versus the number of communication rounds and the number of examples processed in total.

examples seen in Figure 5. This measures the data-efficiency of large-cohort training, and shows that large cohort-training requires significantly more examples per unit-accuracy.

While data-inefficiency also occurs in large-batch training [51], it is especially important in federated learning. Large-cohort training faces greater limitations on parallelizability due to data-sharing constraints. Worse, in realistic cross-device settings client compute times can scale super-linearly with their amount of data, so clients with more data are more likely to become *stragglers* [7]. This straggler effect means that data-inefficient algorithms may require longer training times. To demonstrate this, we show in Appendix B.5 that under the probabilistic straggler runtime model from [38], large-cohort training can require significantly more compute time to converge.

## 4  Diagnosing Large-Cohort Challenges

We now examine the challenges in Section 3, and provide partial explanations for their occurrence. One of the key differences between `FedAvg` and `FedSGD` is what the pseudo-gradient $\Delta$ in (1) represents. In `FedSGD`, $\Delta$ is a stochastic gradient estimate (*i.e.,* $\mathbb{E}[\Delta] = \nabla f$, where the expectation is over all randomness in a given communication round). For special cases of Algorithm 1 where clients perform multiple local training steps, $\Delta$ is not an unbiased estimator of $\nabla f$ [9, 49, 60]. While increasing the cohort size should reduce the variance of $\Delta$ as an estimator of $\mathbb{E}[\Delta]$, it is unclear what this quantity represents.

To better understand $\Delta$, we plot its norm on Stack Overflow in Figures 6a and 6b. For `FedSGD`, $\|\Delta\|$ decreases slightly with $M$, but has high variance. By contrast, for `FedAvg` larger cohorts lead to smaller norms with little overlap. The decrease in norm obeys an inverse square root rule: Let $\Delta_1, \Delta_2$ be pseudo-gradients at some round for cohort sizes $M_1, M_2$. For `FedAvg`, $\|\Delta_1\|/\|\Delta_2\| \approx \sqrt{M_2/M_1}$. We use this rule to predict pseudo-gradient norms for `FedAvg` in Figure 6c. After a small number of rounds, we obtain a remarkably good approximation. To explain this, we plot the average cosine similarity between client updates $\Delta_k^t$ at each round in Figure 6d, with $M = 50$. For `FedAvg`, the client updates are on average almost orthogonal. This explains Figure 6b, as $\Delta$ is an average of nearly orthogonal vectors. As we show in Appendix B.6, similar results hold for other tasks and optimizers.

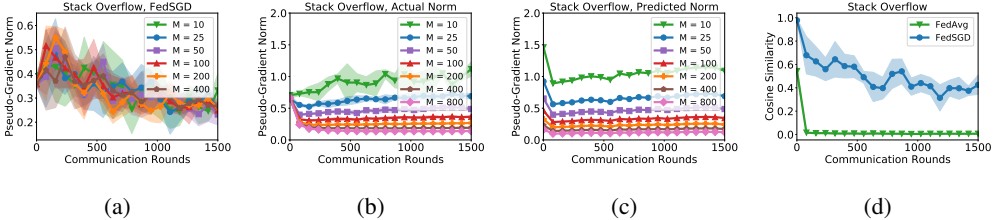

(a)  (b)  (c)  (d)

Figure 6: The pseudo-gradient norm of `FedSGD` (a) and `FedAvg` (b) on Stack Overflow with varying cohort sizes $M$. We also plot the predicted norm for `FedAvg` using an inverse square root scaling rule relative to $M = 50$ (c) and the average cosine similarity of client updates for $M = 50$ (d).

**Implications for large-cohort training.** This near-orthogonality of client updates is key to understanding the challenges in Section 3. The diminishing returns in Section 3.2 occur in part because increasing $M$ leads to smaller updates. This also sheds light on Section 3.5: In large-cohort training, we take an average of many nearly-orthogonal vectors, so each client's examples contribute little. The decreasing pseudo-gradient norms in Figure 6c also highlights an advantage of methods such as

`FedAdam` and `FedAdagrad`: Adaptive server optimizers employ a form of normalization that makes them somewhat scale-invariant, compensating for this norm reduction.

# 5 Designing Better Methods

We now explore an initial set of approaches aimed at improving large-cohort training, drawing inspiration where possible from large-batch training. Our solutions are designed to provide simple baselines for improving large-cohort training. In particular, our methods and experiments are intended to serve as a useful reference for future work in the area, not to fully solve the challenges of large-cohort training.

## 5.1 Learning Rate Scaling

One common technique for large-batch training is to scale the learning rate according to the batch size. Two popular scaling methods are square root scaling [36] and linear scaling [20]. While such techniques have had clear empirical benefit in centralized training, there are many different ways that they could be adapted to federated learning. For example, in Algorithm 1, the client and server optimization both use learning rates that could be scaled.

We consider the following scaling method for large-cohort training: We fix the client learning rate, and scale the server learning rate with the cohort size. Such scaling may improve convergence by compensating for the pseudo-gradient norm reduction in Figure 6. We use square root and linear scaling rules: Given a learning rate $\eta_s$ tuned for $M$, for $M' \geq M$ we use a learning rate $\eta'_s$ where

$$\eta'_s = \frac{\sqrt{M'}}{\sqrt{M}}\eta_s \text{ (square root scaling)} \quad \text{OR} \quad \eta'_s = \frac{M'}{M}\eta_s \text{ (linear scaling).} \tag{3}$$

We also use a version of the warmup strategy from [20]. For the first $W$ communication rounds, we linearly increase the server learning rate from $\eta_s$ to $\eta'_s$. In our experiments, we set $W = 100$ and use a reference server learning rate $\eta_s$ tuned for $M = 50$.

Our experiments show that server learning rate scaling rules have mixed efficacy in large-cohort training. Linear scaling is often too aggressive for federated learning, and caused catastrophic training failures beyond $M = 100$ even when using adaptive clipping (see Appendix B.7). By contrast, square root scaling did not cause catastrophic training failures. Its performance (Figure 7) varied widely across tasks. For example, it significantly improved train accuracy on Shakespeare, but reduced test accuracy. While it led to small accuracy improvements on Stack Overflow for some cohort sies, it degraded accuracy for the largest cohort sizes. In sum, we find that applying learning rate scaling at the server may not directly improve large-cohort training.

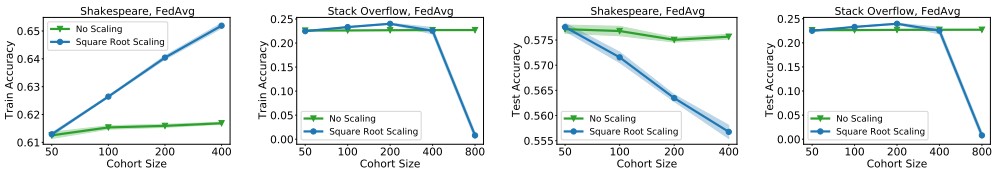

Figure 7: The train and test accuracy of `FedAvg` using square root scaling with warmup, versus no scaling. Results are given for Shakespeare (left) and Stack Overflow (right).

## 5.2 Layer-wise Adaptivity

Another popular technique for large-batch training is *layer-wise adaptivity*. Methods such as LARS [71] and Lamb [72] use layer-wise adaptive learning rates, which may allow the methods to train faster than SGD with linear scaling and warmup in large-batch settings [71, 72]. We propose two new federated versions of these optimizers, `FedLARS` and `FedLamb`. These are special cases of Algorithm 1, where the server uses LARS and Lamb, respectively. Given the difficulties of learning rate scaling above, `FedLARS` and `FedLamb` may perform better in large-cohort settings.

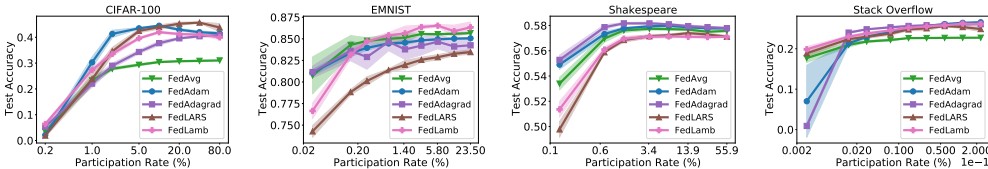

Figure 8: The test accuracy of various methods, including `FedLARS` and `FedLamb`, after training for 1500 rounds, for varying cohort sizes and on varying tasks. The $x$-axis denotes percentage of training clients in each cohort.

In Figure 8 we present the test accuracy of various methods, including `FedLARS` and `FedLamb`, for varying cohort sizes. In most cases, we see that `FedLamb` performs comparably to `FedAdam` for large cohort sizes, but with slightly worse performance in intermediate stages. One notable exception is Stack Overflow, in which `FedLamb` performs well even for $M = 1$. As in Section 3.3, `FedLamb` sees an eventual drop in test accuracy for $M > 100$. `FedLARS` has decidedly mixed performance. While it performs well on CIFAR-10, it does not do well on EMNIST or Shakespeare. While federated layer-wise adaptive algorithms can be better than coordinate-wise adaptive algorithms on certain datasets in some large-cohort settings, our results do not indicate that they are universally better.

## 5.3 Dynamic Cohort Sizes

As we saw in Section 3.5, large-cohort training can reduce data efficiency. Part of this stems from the fact that larger cohorts may help very little for smaller accuracy thresholds (see Figure 5). In order to improve data efficiency, we may be able to use smaller cohorts in earlier optimization stages, and increase the cohort size over time. This technique is parallel to "dynamic batch size" techniques used in large-batch training [66]. In order to test the efficacy of such techniques in large-cohort training, we start with an initial cohort size of $M = 50$ and double the size every 300 rounds up to $M = 800$ (or the maximum population size if smaller). This results in doubling the cohort size a maximum of 4 times over the 1500 rounds of training we perform. We plot the results for `FedAvg` and `FedAdam` on CIFAR-100 and Stack Overflow in Figure 9. See Appendix B.8 for results on all tasks.

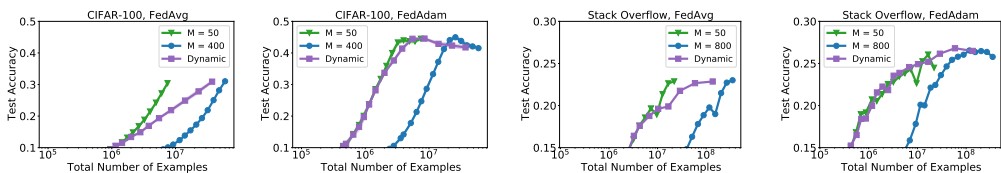

Figure 9: Test accuracy of `FedAvg` and `FedAdam` on Shakespeare (left) and Stack Overflow (right), with respect to the total number of examples processed, using fixed and dynamic cohort sizes.

This dynamic strategy attains data efficiency closer to a fixed cohort size of $M = 50$, while still obtaining a final accuracy closer to having used a large fixed cohort size. While our initial findings are promising, we note two important limitations. First, the accuracy of the dynamic strategy is bounded by the minimum and maximum cohort size used; It never attains a better accuracy than $M = 800$. Second, the doubling strategy still faces the generalization issues discussed in Section 3.3.

## 5.4 Normalized `FedAvg`

While the methods above show promise in resolving some of the issues of large-cohort training, they also introduce extra hyperparameters (such as what type of learning rate scaling to use, or how often to double the cohort size). Hyperparameter tuning can be difficult in federated learning, especially cross-device federated learning [27]. Even adaptive methods like `FedAdam` introduce a number of new hyperparameters that can be challenging to contend with. We are therefore motivated to design a large-cohort training method that does not introduce any new hyperparameters.

Recall that in Section 4, we showed that for `FedAvg`, the client updates ($\Delta_k^t$ in Algorithm 1 and Algorithm 2) are nearly orthogonal in expectation. By averaging nearly orthogonal updates in large-

cohort training, we get a server pseudo-gradient $\Delta^t$ that is close to zero. To compensate, we propose a variant of `FedAvg` where rather than applying SGD to the server pseudo-gradient (as in Algorithm 1), we apply SGD to the normalized server pseudo-gradient. That is, the server updates its model via

$$x' = x - \eta_s \Delta / \|\Delta\|_2.$$

This method, which we refer to as normalized `FedAvg`, is a federated analog of normalized SGD methods used for centralized learning [57]. It introduces no new hyperparameters with respect to Algorithm 1. To test it, we present its training and test accuracy versus cohort size in Figure 10. Notably, we re-use the same learning rates tuned for (unnormalized) `FedAvg`. For full results, see Appendix B.9.

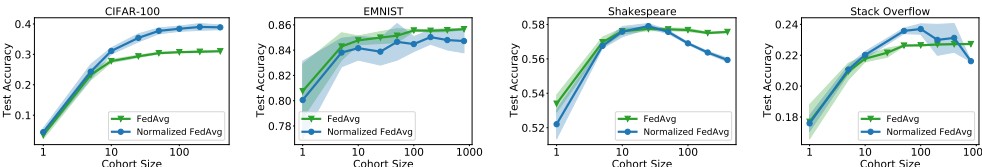

Figure 10: The test accuracy of `FedAvg` and the normalized variant of `FedAvg`, after training for 1500 communication rounds. Results are given for various cohort sizes and tasks.

We find that for most cohort sizes and on most tasks, normalized `FedAvg` achieves better training accuracy for larger cohorts. Thus, this helps mitigate the diminishing returns issue in Section 3.2. We note two important exceptions: for EMNIST, the normalized `FedAvg` is slightly worse for all cohort sizes. For Stack Overflow, it obtains worse training accuracy for the largest cohort size. However, we see significant improvements on CIFAR-100 and all but the largest cohort sizes for Stack Overflow. We believe that the method therefore exhibits promising results, and may be improved in future work.

### 5.5 Hyperparameter tuning and other results.

The methods discussed above, including learning rate scaling and layer-wise adaptivity, can require significant tuning to perform well [58]. To date, little work has been paid to how to tune hyperparameters in federated learning. Such work may be vital to obtain optimal performance, especially given our observations in Section 4 and the client-server structure of federated algorithms, which gives rise to many more hyperparameters. In Algorithm 1, tuning could involve the client optimizer, the client batch size, the server optimizer, and the cohort size. In fact, the client batch size is a key hyperparameter. Recall that clients perform $E$ epochs of mini-batch SGD on their local datasets. Fixing $E$, the batch size dictates the number of local training steps they perform. As we show in Appendix B.10, this number of local steps is critical for achieving maximal performance, and may be necessary to tune according to the cohort size.

## 6 Limitations and Future Work

In this work we explore the benefits and limitations of large-cohort training in federated learning. As discussed in Sections 3.5 and 5, focusing on the number of communication rounds often obscures the data efficiency of a method. This in turn impacts many metrics important to society, such as total energy consumption or total carbon emissions. While we show that large-cohort training can negatively impact such metrics by reducing data-efficiency (see Section 3.5 and Appendix B.5), a more specialized focus on these issues is warranted. Similarly, we believe that an analysis of fairness in large-cohort settings going beyond Section 3.3 would be beneficial.

Future work also involves connecting large-cohort training to other important aspects of federated learning, and continuing to explore connections with growing lines of work in large-batch training. In particular, we wish to see whether noising strategies, especially differential privacy mechanisms, can help overcome the generalization issues of large-cohort training. Personalization may also help mitigate issues of generalization and fairness. Finally, although not a focus of our work, we note that some of the findings above may extend to cross-silo settings, especially if communication restrictions require subsampling clients.

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
