# A Full Experimental Details

## A.1 Datasets and Models

We use four datasets throughout our work: CIFAR-100 [35], the federated extended MNIST dataset (EMNIST) [13], the Shakespeare dataset [8], and the Stack Overflow dataset [3]. The first two datasets are image datasets, the second two are language datasets. All datasets are publicly available. We specifically use the versions available in TensorFlow Federated [4], which gives a federated structure to all four datasets. Below we discuss the specifics of the dataset and classification task, as well as the model used to perform classification.

**CIFAR-100** The CIFAR-100 dataset is a computer vision dataset consisting of $32 \times 32 \times 3$ images with 100 possible labels. While this dataset does not have a natural partition among clients, a federated version was created by Reddi et al. [62] using hierarchical latent Dirichlet allocation to enforce moderate amounts of heterogeneity among clients. This partitioning among clients was based on Pachinko allocation [44]. Note that under this partitioning, each client typically has only a subset of the 100 possible labels. The dataset has 500 training clients and 100 test clients, each with 100 examples in their local dataset.

We train a ResNet-18 [21] on this dataset, where we replace all batch normalization layers with group normalization layers [68]. The use of group norm over batch norm in federated learning was first advocated by Hsieh et al. [24], who showed that this helped improve classification accuracy in the presence of heterogeneous clients. We specifically use group normalization layers with two groups. We perform small amounts of data augmentation and preprocessing for each train and test sample. We first centrally crop each image $(24, 24, 3)$. We then normalize the pixel values according to their mean and standard deviation.

**EMNIST** The EMNIST dataset consists of images hand-written alphanumeric characters. Each image consists of $28 \times 28$ gray-scale pixel values. There are 62 total alphanumeric characters represented in the dataset. The images are partitioned among clients according to their author. The dataset has 3,400 clients, who have both train and test datasets. The dataset has natural heterogeneity stemming from the writing style of each person. We train a convolutional network on the dataset (the same one used by Reddi et al. [62]). The network uses two convolutional layers (each with $3 \times 3$ kernels and strides of length 1), followed by a max pooling layer using dropout with $p = 0.25$, a dense layer with 128 units and dropout with $p = 0.5$, and a final dense output layer.

**Shakespeare** The Shakespeare dataset is derived from the benchmark designed by Caldas et al. [8]. The dataset corpus is the collected works of William Shakespeare, and the clients correspond to roles in Shakespeare's plays with at least two lines of dialogue. To eliminate confusion, *character* here will refer to alphanumeric characters (such as the letter *q*) and symbols such as punctuation, while we will use *client* to denote the various roles in plays (such as Macbeth). There are a total of 715 clients, whose lines are partitioned between train and test datasets.

We split each client's lines into sequences of 80 characters, padding if necessary. We use a vocabulary size of 90, where 86 characters are contained in Shakespeare's work, and the remaining 4 are beginning and end of line tokens, padding tokens, and out-of-vocabulary tokens. We perform next-character prediction on the clients' dialogue using a recurrent neural network (RNN) [55]. We use the same model as Reddi et al. [62]. The RNN takes as input a sequence of 80 characters, embeds it into a learned 8-dimensional space, and passes the embedding through 2 LSTM layers [22], each with 256 units. Finally, we use a softmax output layer with 80 units, where we try to predict a sequence of 80 characters formed by shifting the input sequence over by one. Therefore, our output dimension is $80 \times 90$. We compute loss using cross-entropy loss.

**Stack Overflow** Stack Overflow is a language dataset consisting of question and answers from the Stack Overflow site. The questions and answers also have associated metadata, including tags. Each client corresponds to a user. The specific train/validation/test split from [3] has 342,477 train clients, 38,758 validation clients, and 204,088 test clients. Notably, the train clients only have examples from before 2018-01-01 UTC, while the test clients only have examples from after 2018-01-01 UTC. The validation clients have examples with no date restrictions, and all validation examples are held-out from both the test and train sets.

We perform next-word prediction on this dataset. We restrict each client to the first 1000 sentences in their dataset (if they contain this many, otherwise we use the full dataset). We also perform padding and truncation to ensure that each sentence has 20 words. We then represent the sentence as a sequence of indices corresponding to the 10,000 most frequently used words, as well as indices representing padding, out-of-vocabulary words, the beginning of a sentence, and the end of a sentence. We perform next-word-prediction on these sequences using an a recurrent neural network (RNN) [55] that embeds each word in a sentence into a learned 96-dimensional space. It then feeds the embedded words into a single LSTM layer [22] of hidden dimension 670, followed by a densely connected softmax output layer. Note that this is the same model used by Reddi et al. [62]. The metric used in the main body is the accuracy over the 10,000-word vocabulary; it does not include padding, out-of-vocab, or beginning or end of sentence tokens when computing the accuracy.

## A.2 Implementation and Hyperparameters

We implement the previously proposed methods of `FedAvg`, `FedSGD`, `FedAvgM`, `FedAdam`, `FedAdagrad`, as well as two novel methods, `FedLARS` and `FedLamb`. All implementations are special cases of Algorithm 1. In all cases, clients use mini-batch SGD with batch size $B$. For `FedSGD`, the batch size $B$ of a client is set to the size of its local dataset (so that the client only takes a single step). For all other optimizers, we fix $B$ at a per-task level (see Table 2). Note that we use larger batch sizes for datasets where clients have more examples, like Stack Overflow. Except for the experiments in Appendix B.10, we set $E = 1$ throughout.

Table 2: Batch sizes used for each for all algorithms (except for `FedSGD`) on each dataset.

| DATASET | BATCH SIZE |
|---|---|
| CIFAR-100 | 20 |
| EMNIST | 20 |
| SHAKESPEARE | 4 |
| STACK OVERFLOW | 32 |

For the actual implementation of the algorithms above, all methods (except for `FedSGD`) differ only in the choice of SERVEROPT in Algorithm 1. For `FedSGD`, in addition to having clients use full-batch SGD (as mentioned above), the client learning rate is set to be $\eta_c = 1$ in order to allow Algorithm 1 to recover the version of `FedSGD` proposed by McMahan et al. [52]. For all other algorithms, we present the choice of SERVEROPT and relevant hyperparameters (except for learning rates, see Section A.4) in Table 3. Note that here we use the notation from [33], where $\beta_1$ refers to a first-moment momentum parameter, $\beta_2$ refers to a second-moment momentum parameter, and $\epsilon$ is a numerical stability constant used in adaptive methods. Note that for all adaptive methods, we set their initial accumulators to be 0.

Table 3: Hyperparameters and implementation details for all algorithms, relative to Algorithm 1. Here, $\beta_1$ denotes a first-moment momentum parameter, $\beta_2$ denotes a second-moment momentum parameter, and $\epsilon$ is a value used for numerical stability purposes in adaptive methods.

| ALGORITHM | SERVEROPT | $\beta_1$ | $\beta_2$ | $\epsilon$ |
|---|---|---|---|---|
| `FedAvg` [52] | SGD | 0 | N/A | N/A |
| `FedAvgM` [24] | SGD | 0.9 | N/A | N/A |
| `FedAdagrad` [62] | Adagrad [15] | N/A | N/A | 0.001 |
| `FedAdam` [62] | Adam [33] | 0.9 | 0.99 | 0.001 |
| `FedLARS` | LARS [71] | 0.9 | N/A | 0.001 |
| `FedLamb` | Lamb [72] | 0.9 | 0.99 | 0.001 |

## A.3 Adaptive Clipping

As exemplified in Figure 1, catastrophic training failures can occur when the server pseudo-gradient $\Delta^t$ is too large, which occurs more frequently for larger cohort sizes. To mitigate this issue, we use

---

**Algorithm 2** FedOpt framework with adaptive clipping

---

**Input:** $M, T\ E, x^1, \eta_c, \eta_s, \eta_a, q, \rho^1$, SERVEROPT, $\{p_k\}_{k=1}^K$
**for** $t = 1, \cdots, T$ **do**
    The server selects a cohort $C_t$ of $M$ clients uniformly at random, without replacement.
    The server sends $x^t, \rho^t$ to all clients in $C_t$.
    Each client $k \in C_t$ updates $x^t$ for $E$ epochs of mini-batch SGD with step-size $\eta_c$ on $f_k$.
    After training, each client has a local model $x_k^t$.
    Each client $k \in C_t$ computes $\Delta_k^t = x^t - x_k^t$ and $b_k^t = \mathbb{I}[\|\Delta_k^t\| \le \rho^t]$.
    Each client $k \in C_t$ computes

$$h(\Delta_k^t) = \Delta_k^t \min\left\{1, \frac{\rho^t}{\|\Delta_k^t\|}\right\}.$$

    Each client $k \in C_t$ sends $h(\Delta_k^t)$ and $b_k^t$ to the server.
    The server computes a pseudo-gradient $\Delta^t$ and updates its model via

$$\Delta^t = \frac{\sum_{k \in C_t} p_k h(\Delta_k^t)}{\sum_{k \in C_t} p_k}, \quad x^{t+1} = \text{SERVEROPT}(x_t, \eta_s, \Delta^t).$$

    The server updates its clipping level via

$$b^t = \frac{1}{|C_t|} \sum_{k \in C_t} b_k^t, \quad \rho^{t+1} = \rho^t \exp(-\eta_a(b^t - q)).$$

---

the adaptive clipping method proposed by Andrew et al. [2]. While we encourage the reader to see this paper for full details and motivation, we give a brief overview of the method below.

Recall that in Algorithm 1, $\Delta^t$ is an average of client updates $\Delta_k^t$. Thus, $\Delta^t$ can only be large if some client update is also large. In order to prevent this norm blow-up, we clip the client updates before averaging them. Rather than send $\Delta_k^t$ to the server, for a clipping level $\rho > 0$, the clients send $h(\Delta_k^t, \rho)$ where

$$h(v, \rho) = \begin{cases} v, & \text{if } \|v\| \le \rho \\ \dfrac{\rho v}{\|v\|}, & \text{if } \|v\| > \rho. \end{cases}$$

Instead of fixing $\rho$ a priori, we use the adaptive method proposed by Andrew et al. [2]. In this method, the clipping level varies across rounds, and is adaptively updated via a geometric update rule, where the goal is for $\rho$ to estimate some norm percentile $q \in [0, 1]$. Notably, Andrew et al. [2] show that the clipping level can be learned in a federated manner that is directly compatible with Algorithm 1. At each round $t$, let $\rho^t$ be the clipping level (intended to estimate the $q$th percentile of norms across clients), and let $C_t$ be the cohort of clients selected. Each client $k \in C_t$ computes their local model update $\Delta_k^t$ in the same manner as in Algorithm 1. Instead of sending $\Delta_k^t$ to the server, the client instead sends their clipped update $h(\Delta_k^t, \rho^t)$ to the server, along with $b_k^t := \mathbb{I}[\|\Delta_k^t\| \le \rho^t]$, where $\mathbb{I}[A]$ denotes the indicator function of an event $A$. The server then computes:

$$\Delta^t = \frac{\sum_{k \in C_t} p_k h(\Delta_k^t)}{\sum_{k \in C_t} p_k}, \quad b^t = \frac{1}{|C_t|} \sum_{k \in C_t} b_k^t.$$

That is, $\Delta^t$ is a weighted average of the clipped client updates, and $b^t$ is the fraction of unclipped client updates that did not exceed the clipping threshold. The server then updates its global model as in (2), but it also updates its estimate of the $q$th norm percentile using a learning rate $\eta_a > 0$ via

$$\rho^{t+1} = \rho^t \exp(-\eta_a(b^t - q)). \tag{4}$$

While Andrew et al. [2] add noise in order to ensure that $\rho$ is learned in a differentially private manner, we do not use such noise. Full pseudo-code combining Algorithm 1 and the adaptive clipping mechanisms discussed above is given in Algorithm 2.

Table 4: Server learning rate $\eta_s$ used for each algorithm and dataset.

| ALGORITHM | DATASET | | | |
|---|---|---|---|---|
| | CIFAR-100 | EMNIST | Shakespeare | Stack Overflow |
| FedAvg | 1 | 1 | 1 | 1 |
| FedAvgM | 1 | 1 | 0.1 | 1 |
| FedAdagrad | 0.01 | 0.1 | 0.1 | 10 |
| FedAdam | 0.01 | 0.001 | 0.01 | 1 |
| FedLARS | 0.01 | 0.001 | 0.01 | 0.01 |
| FedLamb | 0.001 | 0.01 | 0.01 | 0.01 |
| FedSGD | 0.1 | 0.1 | 1 | 10 |

Table 5: Client learning rate $\eta_c$ used for each algorithm and dataset.

| ALGORITHM | DATASET | | | |
|---|---|---|---|---|
| | CIFAR-100 | EMNIST | Shakespeare | Stack Overflow |
| FedAvg | 0.1 | 0.1 | 1 | 10 |
| FedAvgM | 0.1 | 0.1 | 1 | 10 |
| FedAdagrad | 0.1 | 0.001 | 10 | 10 |
| FedAdam | 0.1 | 0.1 | 10 | 10 |
| FedLARS | 0.1 | 0.1 | 10 | 1 |
| FedLamb | 0.01 | 0.1 | 10 | 10 |

**Usage and hyperparameters.** We use Algorithm 2 in all experiments (save for those in Figure 1, which illustrate the potential failures that can occur if clipping is not used). For hyperparameters, we use a target percentile of $q = 0.8$, with an initial clipping level of $\rho_1 = 1$. In our geometric update rule, we use a learning rate of $\eta_a = 0.2$.

### A.4 Learning Rates and Tuning

For our experiments, we use client and server learning rates $\eta_s, \eta_c$ that are tuned a priori on a held-out validation dataset. We tune both learning rates over $\{10^i \mid -3 \le i \le 1\}$ for each algorithm and dataset, therefore resulting in 25 possible configurations for each pair. This tuning, like the experiments following it, is based on the algorithm implementations discussed above. In particular, the tuning also uses the adaptive clipping framework discussed in Appendix A.3 and Algorithm 2.

While Stack Overflow has an explicit validation set distinct from the test and train datasets [3], the other three datasets do not. In order to tune on these datasets, we randomly split the training clients (not the training examples!) into train and validation subsets according to an 80-20 split. We then use these federated datasets to perform held-out set tuning. We select the learning rates that have the best average validation performance after 1500 communication rounds with cohort size $M = 10$ over 5 random trials. A table of the resulting learning rates is given in Tables 4 and 5. Note that there is no client learning rate for FedSGD, as we must use $\eta_c = 1$ in Algorithm 1 in order to recover the version of FedSGD in [52]. Note that we use the same learning rates for all cohort sizes.

# B  Full Experiment Results

## B.1  Test Accuracy Versus Communication Round

In this section, we present the test accuracy of various federated learning methods on various tasks, for various cohort sizes. The results are plotted in Figures 11, 12, 13, 14, 15, 16, and 17, which give the results for `FedSGD`, `FedAvg`, `FedAvgM`, `FedAdagrad`, `FedAdam`, `FedLARS`, and `FedLamb` (respectively).

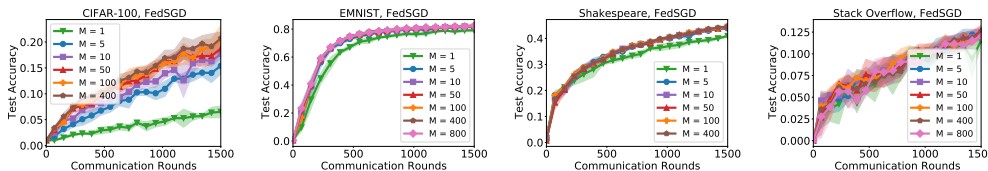

Figure 11: Average test accuracy of `FedSGD` versus the number of communication rounds, for various tasks and cohort sizes $M$.

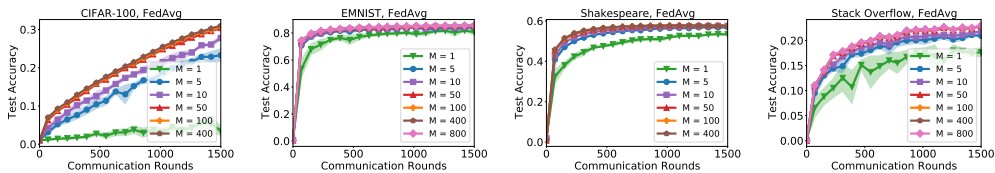

Figure 12: Average test accuracy of `FedAvg` versus the number of communication rounds, for various tasks and cohort sizes $M$.

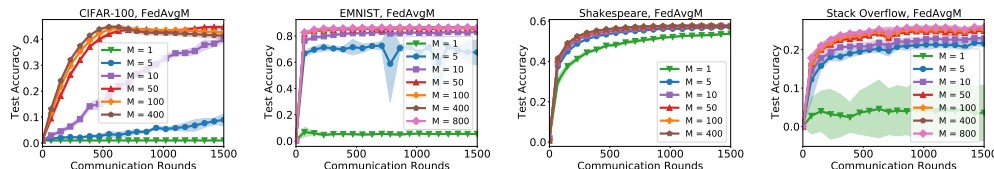

Figure 13: Average test accuracy of `FedAvgM` versus the number of communication rounds, for various tasks and cohort sizes $M$.

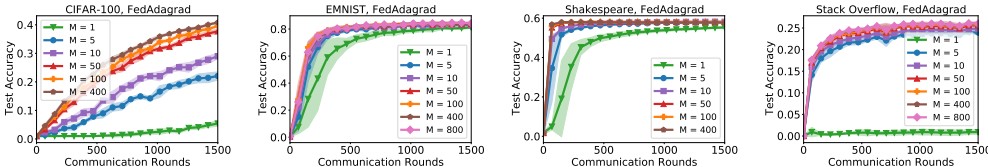

Figure 14: Average test accuracy of `FedAdagrad` versus the number of communication rounds, for various tasks and cohort sizes $M$.

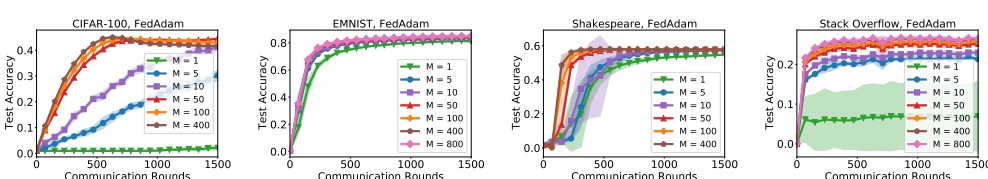

Figure 15: Average test accuracy of `FedAdam` versus the number of communication rounds, for various tasks and cohort sizes $M$.

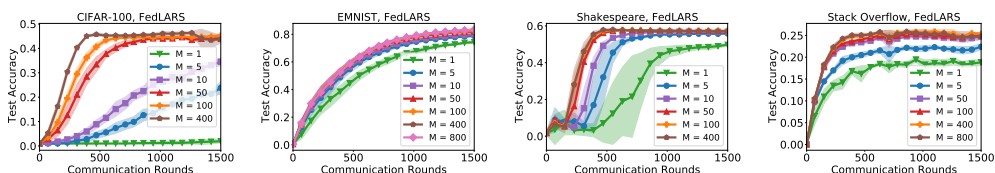

Figure 16: Average test accuracy of `FedLARS` versus the number of communication rounds, for various tasks and cohort sizes $M$.

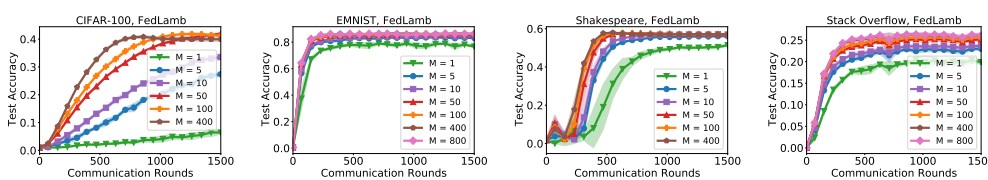

Figure 17: Average test accuracy of `FedLamb` versus the number of communication rounds, for various tasks and cohort sizes $M$.

## B.2  Accuracy Versus Cohort Size

In this section, we showcase the train and test accuracy of various methods, as a function of the cohort size. The results are given in Figures 18 and 19, which correspond to the train and test accuracy, respectively. Both plots give the accuracy of `FedAvg`, `FedAdam`, `FedAdagrad`, `FedLARS`, and `FedLamb` as a function of the *participation rate*. That is, the percentage of training clients used in each cohort.

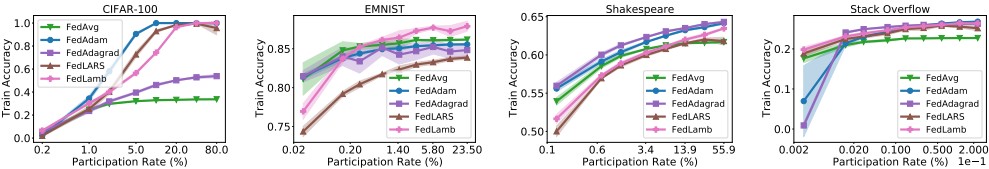

Figure 18: Train accuracy of `FedAvg`, `FedAdam`, `FedAdagrad`, `FedLARS`, and `FedLamb` after 1500 rounds, using varying cohort sizes and tasks. The $x$-axis denotes the percentage of training clients in each cohort.

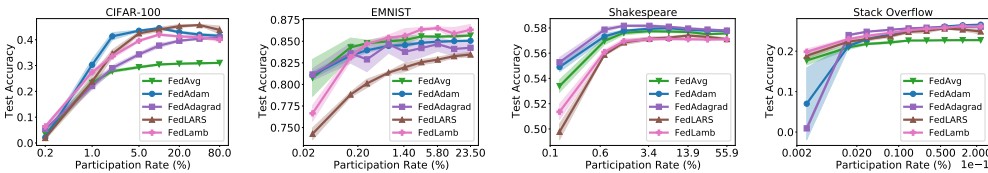

Figure 19: Test accuracy of `FedAvg`, `FedAdam`, `FedAdagrad`, `FedLARS`, and `FedLamb` after 1500 rounds, using varying cohort sizes and tasks. The $x$-axis denotes the percentage of training clients in each cohort.

## B.3  Cohort Size Speedups

In this section, we attempt to see how much increasing the cohort size can speed up a federated algorithm. In particular, we plot the number of rounds needed to obtain a given accuracy threshold versus the cohort size. The results are given in Figures 20, 21, 22, 23, and 24. We see that in just about all cases, the speedups incurred by increasing the cohort size do not scale linearly. That being said, we still see that increasing the cohort size generally always leads to a reduction in the number of rounds needed to obtain a given test accuracy, and can lead to accuracy thresholds unobtainable by small-cohort training in communication-limited settings. While theory shows that in the worst-case, the cohort size leads to linear speedups, we find that this is generally not the case in practice.

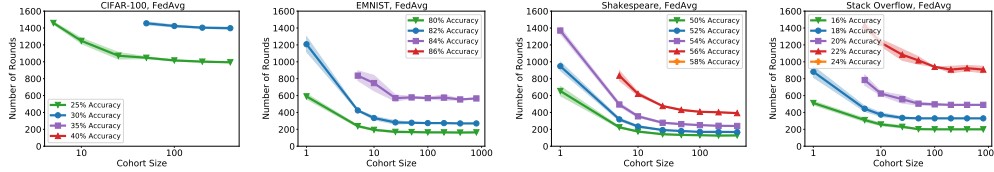

Figure 20: Number of communication rounds for `FedAvg` to obtain certain test accuracy thresholds. The $x$-axis denotes the cohort size.

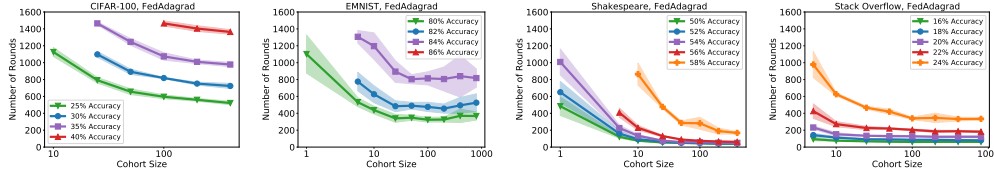

Figure 21: Number of communication rounds for `FedAdagrad` to obtain certain test accuracy thresholds. The $x$-axis denotes the cohort size.

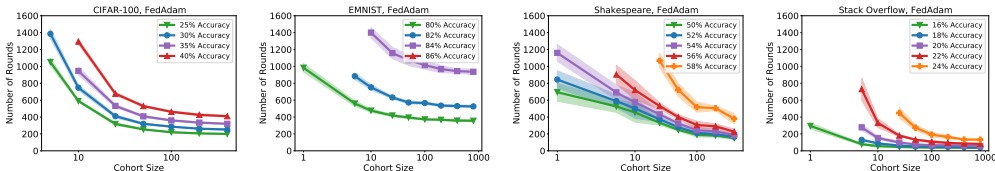

Figure 22: Number of communication rounds for `FedAdam` to obtain certain test accuracy thresholds. The $x$-axis denotes the cohort size.

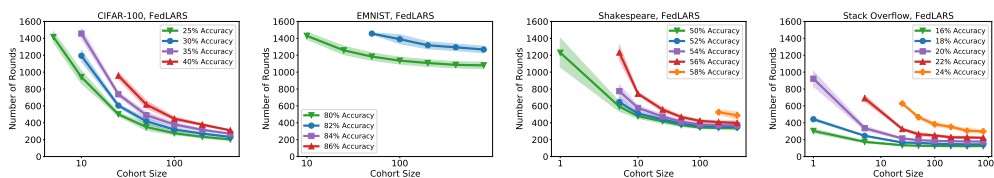

Figure 23: Number of communication rounds for `FedLARS` to obtain certain test accuracy thresholds. The $x$-axis denotes the cohort size.

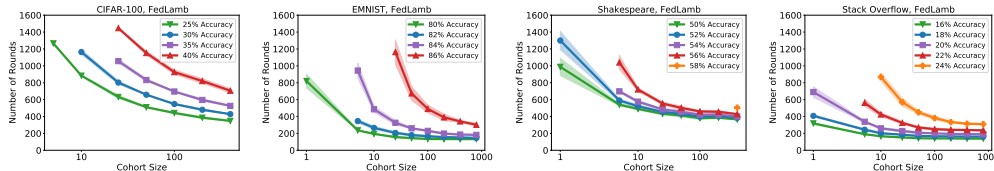

Figure 24: Number of communication rounds for `FedLamb` to obtain certain test accuracy thresholds. The $x$-axis denotes the cohort size.

### B.4 Measures of Accuracy Across Clients

In this section, we expand on the fairness results in Section 3. In Tables 6, 7, 8, and 9, we present percentiles of accuracy of `FedAdam` across all test clients (after training for 1500 rounds, with varying cohort sizes and on varying tasks). For example, the 50th percentile of accuracy is the median accuracy of the learned model across all test clients.

The results show that in nearly all cases, the cohort size impacts all percentiles of accuracy in the same manner. For example, in Table 6, we see that a cohort size of $M = 50$ is better than other cohort sizes, for all percentiles of test accuracy. Notably, this does not support the notion that larger cohorts learn more fair models. Instead, it seems that large cohorts can lead to generalization failures across all percentiles, as it does on CIFAR-100 and Shakespeare (Tables 6 and 8). However, this does not occur on EMNIST and Stack Overflow (Tables 7 and 9), which have many more train and test clients.

Table 6: Percentiles of accuracy across test clients for `FedAdam` on CIFAR-100 after 1500 communication rounds. We present the mean and standard deviation across 5 random trials, with the largest accuracy values for each percentile in bold.

| Percentile | Cohort Size | | | | |
|---|---|---|---|---|---|
| | 10 | 50 | 100 | 200 | 400 |
| 5 | $27.0 \pm 3.3$ | $\mathbf{32.2 \pm 1.1}$ | $30.6 \pm 0.9$ | $29.6 \pm 1.1$ | $29.0 \pm 1.2$ |
| 25 | $35.3 \pm 1.5$ | $\mathbf{39.0 \pm 0.7}$ | $37.5 \pm 0.9$ | $37.2 \pm 1.1$ | $36.4 \pm 1.1$ |
| 50 | $41.1 \pm 1.3$ | $\mathbf{44.5 \pm 0.5}$ | $43.1 \pm 0.7$ | $42.4 \pm 0.5$ | $41.6 \pm 0.7$ |
| 75 | $47.5 \pm 1.1$ | $\mathbf{50.1 \pm 0.7}$ | $48.4 \pm 0.5$ | $47.2 \pm 0.4$ | $47.0 \pm 1.0$ |
| 95 | $54.2 \pm 1.5$ | $\mathbf{55.6 \pm 1.5}$ | $54.6 \pm 1.5$ | $53.8 \pm 1.3$ | $53.6 \pm 1.5$ |

Table 7: Percentiles of accuracy across test clients for `FedAdam` on EMNIST after 1500 communication rounds. We present the mean and standard deviation across 5 random trials, with the largest accuracy values for each percentile in bold.

| Percentile | Cohort Size | | | | | |
|---|---|---|---|---|---|---|
| | 10 | 50 | 100 | 200 | 400 | 800 |
| 5 | $61.9 \pm 2.1$ | $62.5 \pm 2.4$ | $64.3 \pm 1.2$ | $63.8 \pm 1.4$ | $64.3 \pm 1.0$ | $\mathbf{65.0 \pm 0.9}$ |
| 25 | $77.3 \pm 0.7$ | $77.4 \pm 1.4$ | $77.9 \pm 0.4$ | $78.2 \pm 0.4$ | $78.6 \pm 0.5$ | $\mathbf{78.7 \pm 0.3}$ |
| 50 | $84.5 \pm 0.5$ | $85.4 \pm 0.5$ | $85.9 \pm 0.2$ | $85.9 \pm 0.2$ | $86.1 \pm 0.2$ | $\mathbf{86.2 \pm 0.1}$ |
| 75 | $91.2 \pm 0.2$ | $92.0 \pm 0.1$ | $92.0 \pm 0.2$ | $92.3 \pm 0.1$ | $92.3 \pm 0.1$ | $\mathbf{92.3 \pm 0.0}$ |
| 95 | $100.0 \pm 0.0$ | $100.0 \pm 0.0$ | $100.0 \pm 0.0$ | $100.0 \pm 0.0$ | $100.0 \pm 0.0$ | $\mathbf{100.0 \pm 0.0}$ |

Table 8: Percentiles of accuracy across test clients for `FedAdam` on Shakespeare after 1500 communication rounds. We present the mean and standard deviation across 5 random trials, with the largest accuracy values for each percentile in bold.

| Percentile | Cohort Size | | | | |
|---|---|---|---|---|---|
| | 10 | 50 | 100 | 200 | 400 |
| 5 | $\mathbf{39.9 \pm 0.8}$ | $39.2 \pm 1.8$ | $39.4 \pm 1.6$ | $37.8 \pm 0.7$ | $38.6 \pm 1.2$ |
| 25 | $54.9 \pm 0.1$ | $\mathbf{55.0 \pm 0.2}$ | $54.9 \pm 0.2$ | $54.9 \pm 0.1$ | $54.9 \pm 0.2$ |
| 50 | $58.3 \pm 0.2$ | $58.5 \pm 0.2$ | $\mathbf{58.5 \pm 0.2}$ | $58.3 \pm 0.1$ | $58.4 \pm 0.1$ |
| 75 | $61.8 \pm 0.2$ | $\mathbf{62.4 \pm 0.2}$ | $62.2 \pm 0.2$ | $62.1 \pm 0.2$ | $62.1 \pm 0.3$ |
| 95 | $70.9 \pm 0.6$ | $\mathbf{71.2 \pm 0.6}$ | $71.2 \pm 0.4$ | $71.1 \pm 0.2$ | $71.1 \pm 0.2$ |

Table 9: Percentiles of accuracy across test clients for `FedAdam` on Stack Overflow after 1500 communication rounds. We present the mean and standard deviation across 5 random trials, with the largest accuracy values for each percentile in bold.

| Percentile | Cohort Size | | | | | |
|---|---|---|---|---|---|---|
| | 10 | 50 | 100 | 200 | 400 | 800 |
| 5 | $16.7 \pm 0.2$ | $18.7 \pm 0.2$ | $19.1 \pm 0.2$ | $19.5 \pm 0.1$ | $19.8 \pm 0.1$ | $\mathbf{19.9 \pm 0.1}$ |
| 25 | $21.0 \pm 0.3$ | $23.2 \pm 0.2$ | $23.6 \pm 0.2$ | $24.1 \pm 0.1$ | $24.4 \pm 0.1$ | $\mathbf{24.5 \pm 0.1}$ |
| 50 | $23.5 \pm 0.3$ | $25.8 \pm 0.2$ | $26.3 \pm 0.3$ | $26.7 \pm 0.1$ | $27.0 \pm 0.1$ | $\mathbf{27.2 \pm 0.1}$ |
| 75 | $26.1 \pm 0.3$ | $28.4 \pm 0.2$ | $29.0 \pm 0.3$ | $29.4 \pm 0.1$ | $29.7 \pm 0.1$ | $\mathbf{29.9 \pm 0.1}$ |
| 95 | $30.8 \pm 0.3$ | $33.2 \pm 0.2$ | $33.7 \pm 0.4$ | $34.2 \pm 0.1$ | $34.6 \pm 0.1$ | $\mathbf{34.8 \pm 0.1}$ |

## B.5 Simulating Straggler Effects

As shown in Section 3.5, large-cohort training methods seem to face data-efficiency issues, where training with large cohorts requires processing many more examples to reach accuracy thresholds than small-cohort training. While this is related to diminishing returns (Section 3.2) and occurs in large-batch training as well [19], we highlight this issue due to its consequences in federated learning.

Unlike centralized learning, federated learning faces fundamental limits on parallelization. Since data cannot be shared, we typically cannot scale up to arbitrarily large cohort sizes. Instead, the parallelization is limited by the available training clients. In order to learn on a client's local dataset,

that client must actually perform the training on its examples. Unfortunately, since clients are often lightweight in cross-device settings [27], clients with many examples may require longer compute times, becoming stragglers in a given communication round. If an algorithm is data-inefficient, these straggler clients may have to participate many times throughout training, causing the overall runtime to be greater. In short, data inefficiency can dramatically slow down large-cohort training algorithms.

To exemplify this, we compute simulated runtimes of federated algorithms under a version of the probabilistic straggler model from [38]. We model each client's runtime as a random variable drawn from a shifted exponential distribution. Such models were found to be good models of runtimes for file queries in cloud storage systems [45] and mini-batch SGD on distributed compute systems [38].

In our model, we assume that the time a client requires to perform local training is some constant proportional to the number of examples the client has, plus an exponential random variable. More formally, let $N_k$ denote the number of examples held by some client $k$, and let $X_k$ denote the amount of time required by client $k$ to perform their local training in Algorithm 1. Then we assume that there are constants $\alpha, \lambda > 0$ such that

$$ X_k - \alpha N_k \sim \text{Exp}\left(\frac{1}{\lambda N_k}\right). $$

Here $\lambda$ is the *straggler parameter*. Recall that if $X \sim \text{Exp}(1/\lambda)$, then $\mathbb{E}[X] = \lambda$. Therefore, we assume that the expected runtime of client $k$ equal $\alpha N$ plus some random variable whose expected value is $\lambda N$. Thus, larger $\lambda$ means larger expected client runtimes. By convention, we can also use $\lambda = 0$ in which case $X_k = \alpha N_k$. For a given round $t$ of Algorithm 1, let $C_t$ denote the cohort sampled. Since Algorithm 1 requires all clients to finish before updating its global model, we model the runtime $Y_t$ of round $t$ as

$$ Y_t = \max_{k \in C_t} \{X_k\}. $$

Thus, the round runtime is the maximum of $M$ shifted exponential random variables, where $M$ is the cohort size. Note that this only models the client computation time, not the server computation time or communication time. Using this model, we plot the simulated runtime of `FedAvg` on various tasks, for varying cohort sizes. For simplicity, we assume $\alpha = 1$ in all experiments, and vary $\lambda$ over $\{0.1, 1, 10, 100\}$. To showcase how much longer the runtime of large-cohort training may be, we present the simulated runtime, *relative* to $M = 10$. For $a \in [0, 1]$, we plot the ratio of how long it takes to reach a test accuracy of $a$ with a cohort size of $M$, versus how long it takes to reach $a$ with $M = 10$. We give the results for CIFAR-100, EMNIST, Shakespeare, and Stack Overflow in Figures 25, 26, 27, 28, respectively.

When $\lambda$ is small, we see that larger cohorts can obtain higher test accuracy in a comparable amount of time to $M = 10$. However, when $\lambda$ is large, large-cohort training may require anywhere from 5-10 times more client compute time. This is particularly important in cross-device settings with lightweight edge devices, as the straggler effect (which essentially increases with $\lambda$) may be larger. Note that we see particularly large increases in relative runtimes for smaller accuracy thresholds, which suggests that the dynamic cohort strategy from Section 5 may be useful in helping mitigate such issues.

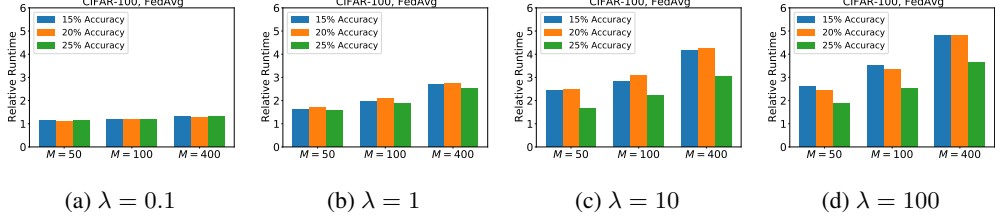

Figure 25: The relative amount of time required to reach given test accuracies on CIFAR-100 with varying cohort sizes. We present the ratio of the runtime needed for $M > 10$ with respect to the time needed for $M = 10$. Runtimes are simulated under a shifted exponential model with $\alpha = 1$ and varying $\lambda$.

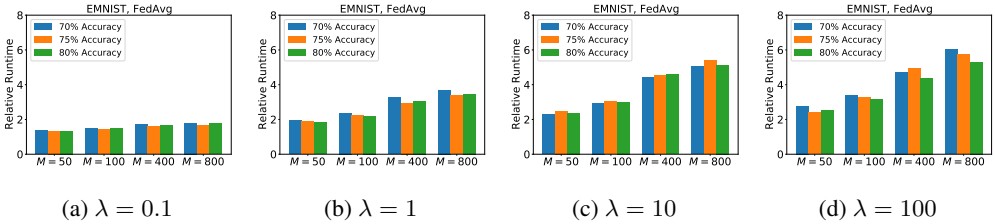

Figure 26: The relative amount of time required to reach given test accuracies on EMNIST with varying cohort sizes. We present the ratio of the runtime needed for $M > 10$ with respect to the time needed for $M = 10$. Runtimes are simulated under a shifted exponential model with $\alpha = 1$ and varying $\lambda$.

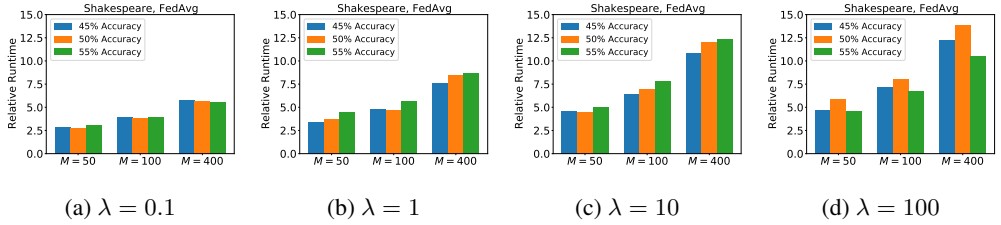

Figure 27: The relative amount of time required to reach given test accuracies on Shakespeare with varying cohort sizes. We present the ratio of the runtime needed for $M > 10$ with respect to the time needed for $M = 10$. Runtimes are simulated under a shifted exponential model with $\alpha = 1$ and varying $\lambda$.

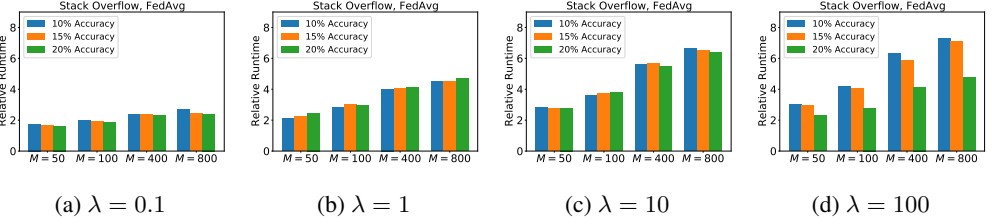

Figure 28: The relative amount of time required to reach given test accuracies on Stack Overflow with varying cohort sizes. We present the ratio of the runtime needed for $M > 10$ with respect to the time needed for $M = 10$. Runtimes are simulated under a shifted exponential model with $\alpha = 1$ and varying $\lambda$.

### B.6 Pseudo-Gradient Norms

In this section, we present the norm of the server pseudo-gradient $\Delta$ in Algorithm 1 with respect to the number of communication rounds. We do this for varying cohort sizes and tasks across 1500 communication rounds. All plots give the $\ell_2$ norm of $\Delta$. The results are given in Figures 29, 30, 31, 32, 33, 34, and 35. These gives the results for FedSGD, FedAvg, FedAvgM, FedAdagrad, FedAdam, FedLARS, and FedLamb (respectively).

We find that in nearly all cases, the results for FedSGD differ from all other algorithms. While there is significant overlap in the pseudo-gradient norm for FedSGD across all cohort sizes (Figure 29), any method that uses multiple local training steps generally does not see such behavior. The only notable counter-example is FedAdagrad on EMNIST (Figure 32). Otherwise, both non-adaptive and adaptive federated methods that use local training (such as FedAvg, FedAdam, and FedLamb) see similar behavior: The pseudo-gradient norm is effectively stratified by the cohort size. Larger cohort sizes lead to smaller pseudo-gradient norms, with little overlap. Moreover, as discussed in Section 4, we see that after enough communication rounds occur, the pseudo-gradient norm obeys an inverse square root scaling rule.

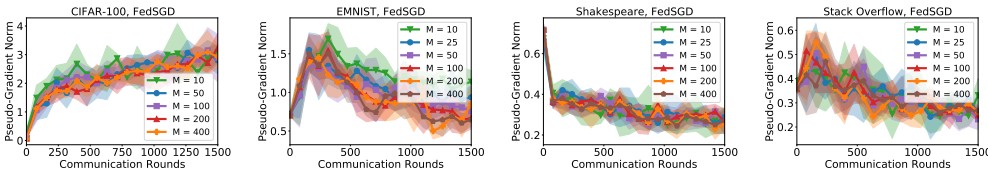

Figure 29: Average pseudo-gradient norm of `FedSGD` versus the number of communication rounds, for various tasks and cohort sizes $M$.

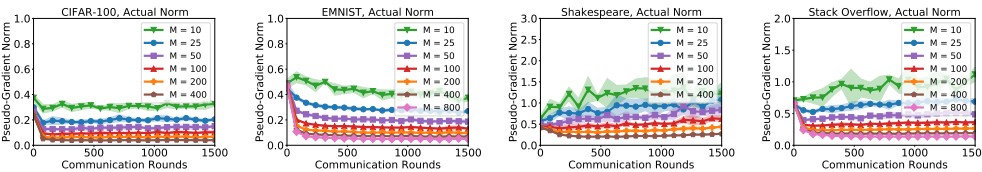

Figure 30: Average pseudo-gradient norm of `FedAvg` versus the number of communication rounds, for various tasks and cohort sizes $M$.

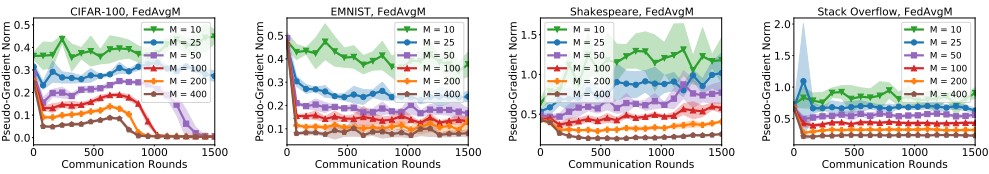

Figure 31: Average pseudo-gradient norm of `FedAvgM` versus the number of communication rounds, for various tasks and cohort sizes $M$.

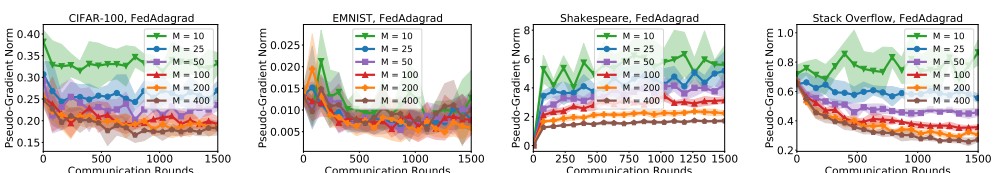

Figure 32: Average pseudo-gradient norm of `FedAdagrad` versus the number of communication rounds, for various tasks and cohort sizes $M$.

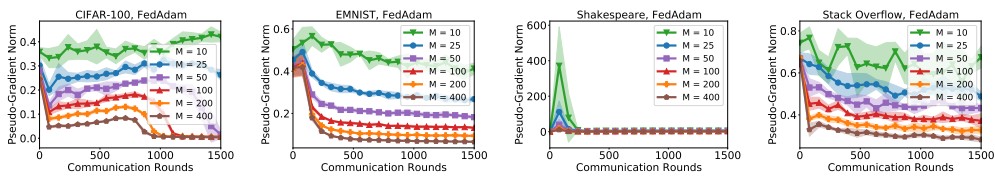

Figure 33: Average pseudo-gradient norm of `FedAdam` versus the number of communication rounds, for various tasks and cohort sizes $M$.

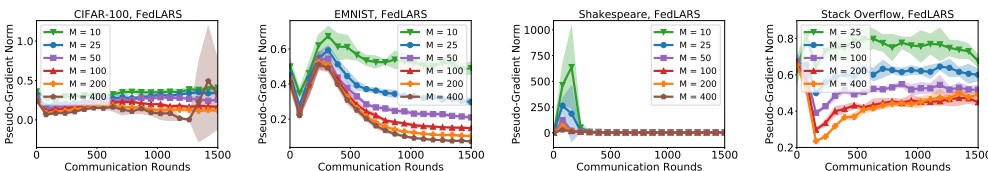

Figure 34: Average pseudo-gradient norm of `FedLARS` versus the number of communication rounds, for various tasks and cohort sizes $M$.

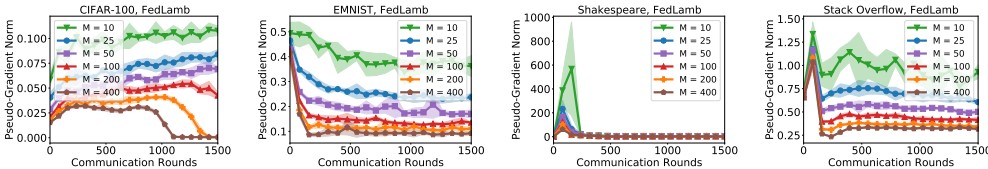

Figure 35: Average pseudo-gradient norm of `FedLamb` versus the number of communication rounds, for various tasks and cohort sizes $M$.

### B.6.1 Cosine Similarity of Client Updates

Recall that in Section 4, we showed that for `FedAvg`, client updates are nearly orthogonal on the Stack Overflow task. In this section, we show that this holds across tasks. In Figure 36, we present the average cosine similarity between distinct clients in each training round, for `FedAvg` and `FedSGD`. Thus, given a cohort size $M$, at each round $t$ we compute $\binom{|M|}{2}$ cosine similarities between client updates, and take the average over all pairs. Formally, we compute, for each round $t$,

$$\theta_t := \binom{|C_t|}{2}^{-1} \sum_{\substack{i,j \in C_t \\ i \neq j}} \frac{\langle \Delta_i^t, \Delta_j^t \rangle}{\|\Delta_i^t\|_2 \|\Delta_j^t\|_2} \tag{5}$$

where $C_t$ is the cohort of sampled clients in round $t$, and $\Delta_k^t$ denotes the client update of client $k \in C_t$ (see Algorithm 1). Note that because we normalize, it does not matter whether we use clipping or not (Algorithm 2). The results for $\theta_t$ with cohort size $M = 50$ are given in Figure 36.

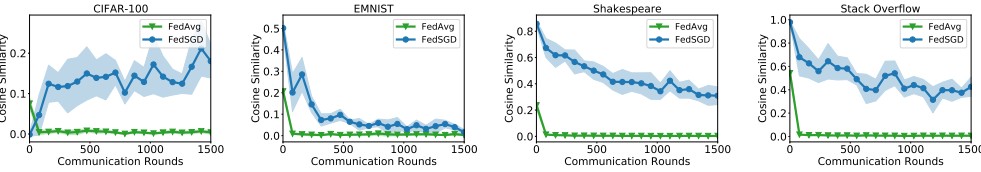

Figure 36: Average cosine similarity $\theta_t$ (as in (5)) between client updates $\Delta_k^t$ with respect to the number of communication rounds, for `FedAvg` on EMNIST with a cohort size of $M = 50$.

We see that in all cases, after a small number of communication rounds, $\theta_t$ becomes close to zero for `FedAvg`. By contrast, $\theta_t$ is not nearly as small for `FedSGD`, especially in intermediate rounds. We note that for EMNIST, the cosine similarity for `FedSGD` approaches that of `FedAvg` as $T \to 1500$.

### B.7 Server Learning Rate Scaling

In this section, we present our full results using the learning rate scaling methods proposed in Section 5. Recall that our methods increase the server learning rate $\eta_s$ in accordance with the cohort size. To do so, we fix a learning rate $\eta_s$ for some cohort size $M$. As in (3), for $M' \geq M$, we use a server learning rate $\eta_s'$

$$\eta_s' = r\left(\frac{M'}{M}\right)\eta_s$$

where $r : \mathbb{R}_{\geq 0} \to \mathbb{R}_{\geq 0}$ determines the scaling rate. In particular, we focus on $r(a) = \sqrt{a}$ (square root scaling) and $r(a) = a$ (linear scaling). These rules both can be viewed as federated analogs of learning rate scaling techniques used for large-batch training [20, 36]. We use them with a federated version of the warmup technique proposed by Goyal et al. [20], where we linearly increase the server learning rate from 0 to $\eta_s'$ over the first $W = 100$ communication rounds.

Despite the historical precedent for the linear scaling rule [20], we find that it leads to catastrophic training failures in the federated regime, even with adaptive clipping. To showcase this, we plot the accuracy of `FedAvg` on EMNIST with the linear scaling rule in Figure 37. We plot the test accuracy over time, averaged across 5 random trials, for various cohort sizes $M$. While $M = 50, 100$ see

similar convergence as in Figure 12, for $M = 200$, we saw one catastrophic training failure across all 5 trials. Using $M \geq 400$, we found that all trials resulted in catastrophic training failures. In short, linear scaling can be too aggressive in federated settings, potentially due to heterogeneity among clients (which intuitively requires some amount of conservatism in server model updates).

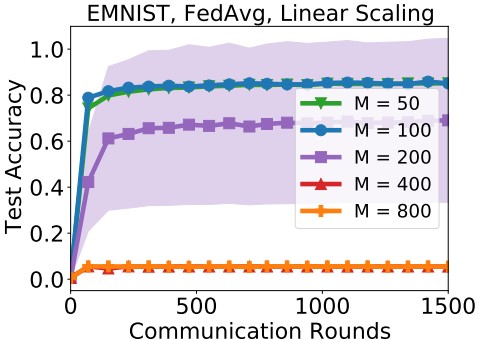

Figure 37: The test accuracy of `FedAvg` on EMNIST with the linear scaling rule, across 5 random trials. The mean accuracy is given in bold, with the standard deviation indicated by the pale region. We see that for $M = 200$, there are a number of catastrophic training failures, while for $M \geq 400$, all trials experienced catastrophic training failures.

By contrast, the square root scaling rule did not lead to such training failures. We plot the training accuracy and test accuracy of `FedAvg` using the square root scaling rule in Figure 38. We plot this with respect to the cohort size, with and without the scaling rule. We see that the performance of the scaling rule is decidedly mixed. While it leads to significant improvements in training accuracy for CIFAR-100 and Shakespeare, it leads to only minor improvements (or a degradation in training accuracy) for EMNIST and Stack Overflow. Notably, while the training accuracy improvement also led to a test accuracy improvement for CIFAR-100, the same is not true for Shakespeare. In fact, the training benefits of the square root scaling there led to worse generalization across the board.

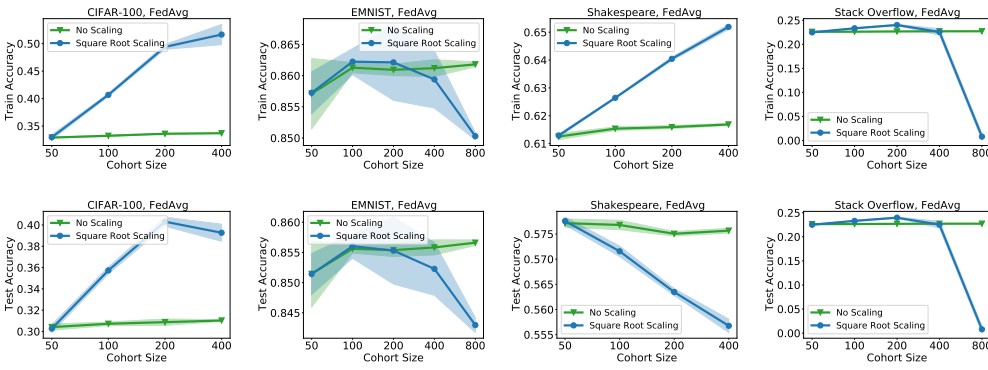

Figure 38: The train accuracy (top) and test accuracy (bottom) of `FedAvg` using square root scaling with warmup, versus no scaling, after training for 1500 communication rounds. Results are given for various cohort sizes and tasks

## B.8 Dynamic Cohort Sizes

In this section, we plot the full results of using the dynamic cohort size strategy from Section 5. Recall that there, we use an analog of dynamic batch size methods for centralized learning, where the cohort size is increased over time. We specifically start with a cohort size of $M = 50$, and double every 300 communication rounds. If doubling would ever make the cohort size larger than the number of training clients, we simply use the full set of training clients in a cohort.

We plot the test accuracy of `FedAdam` and `FedAvg` using the dynamic cohort size, as well as fixed cohort sizes of $M = 50$ and $M = 400$ (for CIFAR-100 and Shakespeare) or $M = 800$. In Figure 39, the test accuracy is plotted with respect to the number of examples processed by the clients, in order to measure the data-efficiency of the various methods. We find that while the dynamic cohort strategy can help interpolate the data efficiency between small and large cohort sizes, obtaining the same data efficiency as $M = 50$ for most accuracy thresholds, then transitioning to the data efficiency of larger $M$.

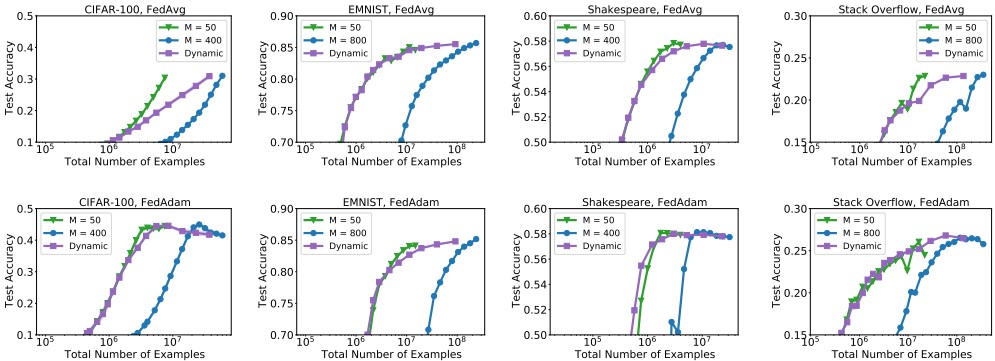

Figure 39: Test accuracy of `FedAvg` (top) and `FedAdam` (bottom) with respect to the total number of examples. Both algorithms are applied to various tasks, with various fixed cohort sizes, and the dynamically increasing cohort strategy.

In Figure 40, we plot the test accuracy of the methods discussed above with respect to the number of communication rounds, in order to better visualize the generalization behavior of the dynamic cohort strategy. We see that for `FedAvg`, there is little to no difference between the test accuracy for $M = 50$ and $M = 400$ or $M = 800$, and that the dynamic cohort strategy generally lays in-between these two. This is partially a consequence of the diminishing returns discussed in Section 3.2. For `FedAdam`, we see that there are more returns to be had for increasing the cohort size. Moreover, we see that the dynamic cohort strategy typically begins at the accuracy level of $M = 50$, and later matches that of the larger cohort. This can be beneficial such as in the case of Stack Overflow, or it can be detrimental as in the case of CIFAR-100, where we see that the dynamic cohort strategy faces the generalization issues in Section 3.3. Thus, we see that the dynamic cohort strategy can help improve the data efficiency of large cohort training, but cannot remedy issues of diminishing returns or generalization failures.

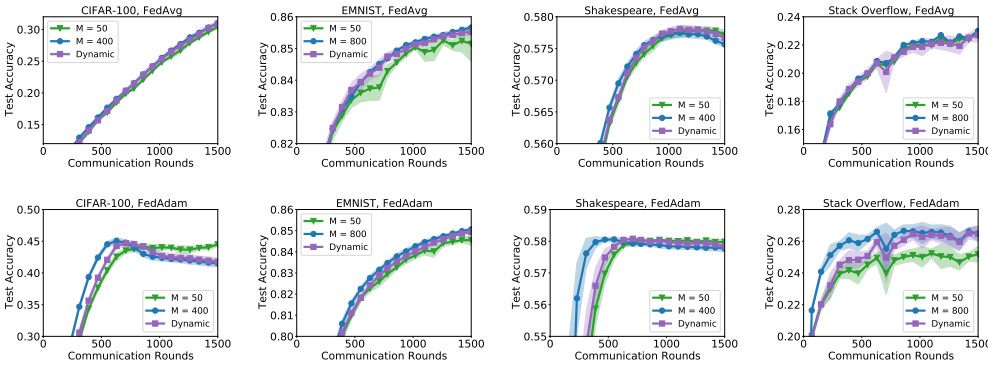

Figure 40: Test accuracy of `FedAvg` (top) and `FedAdam` (bottom) with respect to the total number of communication rounds. Both algorithms are applied to various tasks, with various fixed cohort sizes, and the dynamically increasing cohort strategy.

## B.9 Normalized `FedAvg`

Throughout Section 3, we saw that `FedAvg` faces a number of challenges when performing large-cohort training, including diminishing returns. In Section 4 and Appendix B.6.1, we showed that part of what makes `FedAvg` (with respect to `FedSGD`) are the statistical properties of its training dynamics: In methods such as `FedAvg` (and more generally, algorithms that use multiple client update steps), the client updates ($\Delta_k^t$ in Algorithm 1) are nearly orthogonal. This in turn suggests a partial explanation for the challenges in Section 3. By averaging nearly orthogonal updates in large-cohort training, we get a server pseudo-gradient ($\Delta^t$ in Algorithm 1) that is close to zero in norm (Appendix B.6.1). This in turn means that the server does not make much progress at each communication round.

One straightforward solution would be to simply scale up the server learning rate. As we show in Appendix B.7, this may not result in better training performance across all tasks. Moreover, some server learning rate scaling techniques (such as linear scaling) can cause catastrophic training failures. Such scaling strategies also introduce extra hyperparameters concerning how much scaling should occur. In order to avoid these pitfalls, we propose a variant of `FedAvg` (normalized `FedAvg`) which scales up the pseudo-gradient directly. That is, the server updates its model via

$$x' = x - \eta_s \frac{\Delta}{\|\Delta\|_2}.$$

Note that this introduces no new hyperparameters with respect to Algorithm 1. To test this method, we compare it with unnormalized `FedAvg` across all tasks and multiple cohort sizes. Notably, we do not re-tune any learning rates. We simply use the same learning rates tuned for unnormalized `FedAvg`. The results are given in Figure 41.

We find that for most cohort sizes and tasks, normalized `FedAvg` attains better training accuracy than `FedAvg` for larger cohorts. There are two notable exceptions. For EMNIST, normalized `FedAvg` is slightly worse for all cohort sizes. For Stack Overflow, normalized `FedAvg` obtained worse train accuracy for the largest cohort size. We also see that normalized `FedAvg` sees varying generalization behavior across tasks. While the training benefits for CIFAR-100 translated to improved generalization, the same is not true for Shakespeare. For Stack Overflow, we see that the test accuracy mirrors the training accuracy, so that normalized `FedAvg` improves test accuracy for $M \leq 400$ but degrades test accuracy for $M = 800$. While normalized `FedAvg` did not lead to improvements uniformly, we found that it achieved similar behavior to square root server learning rate scaling (Figure 38), without introducing new hyperparameters or requiring re-tuned learning rates. We believe the method therefore exhibits promise, and may be improved in future work.

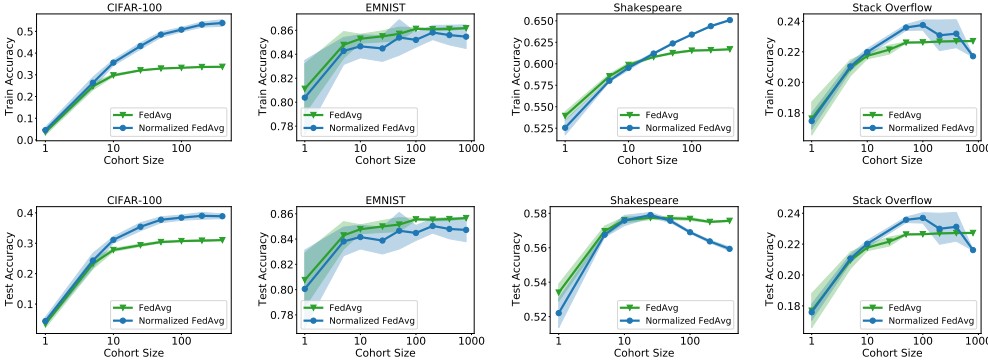

Figure 41: The train accuracy (top) and test accuracy (bottom) of `FedAvg` and the normalized variant of `FedAvg`, after training for 1500 communication rounds. Results are given for various cohort sizes and tasks.

## B.10 Changing the Number of Local Steps

Cohort size is not the only factor determining the number of examples seen per round in Algorithm 1. The number of client epochs $E$ and the client batch size also affect this. To study this "effective batch size" in FL, we fix the client batch size, and investigate how the cohort size and number of local steps

simultaneously impact the performance of `FedAvg`. We fix a local batch size of 1 and vary the cohort size over $\{16, 32, \ldots, 1024\}$. We vary the number of local steps over $\{1, 2, 4, \ldots, 256\}$. We plot the number of rounds needed for convergence, and the final test accuracy in Figure 42. By construction, each square on an anti-diagonal corresponds to the same number of examples per round.

In the left figure, we see that if we fix the cohort size, then increasing the number of local steps can accelerate convergence, but only up to a point, after which catastrophic training failures occur. By contrast, if we have convergence for some number of local steps and cohort size, convergence occurs for all cohort sizes. Similarly, we see in the right hand figure that increasing the number of local steps can drastically reduce generalization, more so than increasing the cohort size. In essence, we see that the number of local steps obeys many of the same issues outlined in Section 3. Therefore, correctly tuning the number of local steps in unison with the cohort size may be critical to ensuring good performance of large-cohort methods.

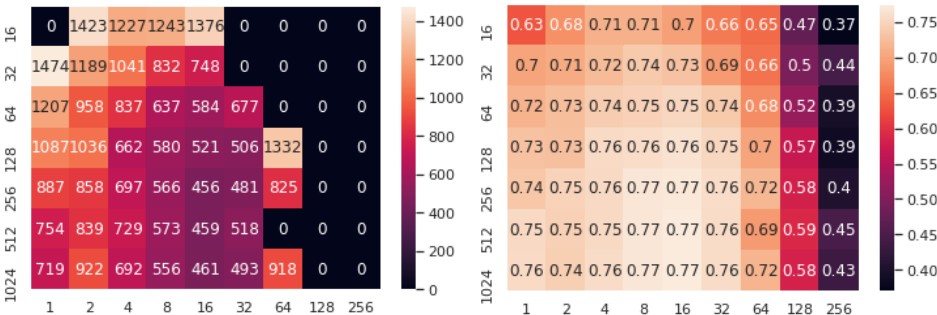

Figure 42: The number of rounds for to reach a test accuracy of $70\%$ (left) and the test accuracy after 1500 rounds (right). Results are for `FedAvg` on EMNIST with varying numbers of local steps ($x$-axis) and cohort sizes ($y$-axis).