# OpenReview forum: "On Large-Cohort Training for Federated Learning"
_NeurIPS.cc/2021/Conference — NeurIPS 2021 Poster_

### Official Review · Reviewer_SF2P · 2021-07-15

**Rating:** 5
**Confidence:** 4

**Summary:**

The authors provide a comprehensive empirical evaluation of the effect of the cohort size (i.e., the number of clients sampled in each round) on the convergence and quality of the model learnt by federated learning algorithms. They investigate different versions of the FedOpt algorithm on 5 different model, dataset combinations. Similar to large mini-batch training in centralized settings they find that generalization properties can suffer from large cohort sizes, convergence speedups from increasing the cohort size are diminishing and the algorithm suffers worse data efficiency. In addition, specific to heterogeneous data settings, they show that larger cohort sizes increase the risk of divergence due to extreme outlier gradients.

Towards addressing these challenges the authors propose to use gradient clipping for federated learning to achieve robustness to unbounded gradients, and they propose to consider layer-wise adaptive algorithms such as LARS or Lamb for server-side aggregation.

Finally, they investigate the sources for these phenomena and show that gradients of individual agents are often orthogonal which they think could partially explain the diminishing return phenomena, the data inefficiency, and the advantage of adaptive step size schedules for large cohort training.


**Limitations And Societal Impact:**

yes

**Main Review:**

The paper is well written, the setup is clear and easy to follow. I like the selection of datasets and tasks. Figures are nice, although they could be a bit larger and fill the entire column.

I think the first observation that gradient clipping is important in federated learning to achieve robustness is interesting. I had not seen it being discussed previously. One question I have here is – how often do you observe these convergence issues? Does it also happen on other datasets, or is it just one particular client in the EMNIST training set that performs poorly? In any case I don’t see a reason why this should not be implemented as a default robustness feature in FL algorithms.

Related to the other claims of the paper I have a couple of questions and concerns that I hope the authors can clarify or comment on.

1) You say that the failure rate increases from 0% to 80% when increasing the cohort size from 10 to 800. Is this evaluated for a fixed number of communication rounds, or for a target accuracy level? If it is a fixed number of rounds is there more going on here than the fact that if there is a small number of bad clients in the training set the probability of sampling it in the course of the algorithm is 80x higher in the larger cohort? (0% for 5 reps could as well be 1%).
2) In L190 you state that the generalization failure do not occur on EMNIST and Stack Overflow because the cohort size is small compared to the total number of clients. I am not sure I understand this reasoning, you are picking the cohort size in your experiment. If you choose the participation on these tasks to 80% can you observe these effects too?
4) I think it is interesting to visualize how the test accuracy is distributed across the clients in the test set, but I don’t think any conclusion about fairness issues arising from the cohort size is justified from your results. I would rather say that small cohort sizes have not yet converged after 1500 rounds, and that’s why the spread in accuracy values is smaller. For a fair comparison I think you would need to compare the distributions for different cohort sizes at a fixed test accuracy.
5) I don’t understand the reasoning in L220. You say that clients with more data are more likely to become stragglers and thus large-cohort training requires more compute time. Isn’t the amount of data per client independent from the cohort size? I would rather think that the probability of having a straggler is larger with more parallel updates and this can increase runtime, but I don’t see the connection to the amount of local data.
6) I don’t agree with the interpretation of the results in section 4. It is obvious from your results that increasing M leads to smaller updates. But I would argue that the true gradient of the training objective is very small (i.e. the heterogeneous updates from the clients approximately sum to zero which is to be expected once the algorithm is converging) and the norm of the gradient you plot is larger for small cohort sizes due to the variance in the individual terms of the sum. Your figures show that even after the algorithm has converged the norm of the update is almost constant and what we are left with is the fluctuations due to heterogeneity in the objectives. Do you think there is more going on here with the inverse square root rule you propose, or is it just reflecting this variance term?
7) You say that the learning rate adjustment has mixed efficacy because the learning rate schedule is often too aggressive and you conclude that applying learning rate scaling at the server may not directly improve large-cohort training. Have you tried tuning the learning rate individually for the different cohort sizes to see if it is an issue of the schedule or whether there is no real potential in tuning the server-side learning rate?
8) You talk about large cohort sizes, and that it might be good to dynamically increase it over time. I think it would be helpful to emphasize that this setting is quite different from mini-batch training because, as you said in the motivation, you cannot control client’s availability in federated learning. It seems that a more natural question would be to ask how you can dynamically react to a changing number of available clients.

Overall, I appreciate the effort of the authors to conduct this extensive study. However, currently there are too many claims and conclusions in the paper that I am not convinced by. If the authors can pursuade me otherwise I am happy to increase my score.


**Time Spent Reviewing:**

4

---

> ### Author Response · Authors · 2021-08-10
> **Response to Reviewer SF2P**
>
> We thank the reviewer for their time and effort, and for their valuable suggestions and questions. Responses to the various points and questions raised are below.
>
> **Regarding the catastrophic training failures:** These occur on all datasets, with all algorithms. Moreover, it is not the case that there is a single “bad” client. It is more subtle, as the failures only occur after FedAvg has seen the client many times (in particular, the failure always seems to arise after sampling certain clients once the model has sufficiently converged). We will clarify this.
>
> **Response to the numbered questions:**
>
> 1. The failure rates were evaluated for a fixed number of training rounds. However, we frequently had experiments where we converged even when seeing the “misaligned” clients (in fact, they were seen multiple times). For example, the failure rate with M = 400 was only 40%, not 80%, but in both cases the cohort size was large enough to sample all clients multiple times throughout training, and to converge. We will note this more clearly in our revision.
> 2. We did not use higher participation rates for these datasets because they have (respectively) 3,400 and 342,477 training clients. Thus, not only is it prohibitively expensive to do simulations at these scales (especially Stack Overflow), it also does not represent cross-device learning, which is a focal point of our paper and is characterized by partial participation (see Kairouz et al., 2019, Table 1). That being said, we were able to do some experiments on EMNIST with a subset of the optimizers, but with 100% participation. We found no generalization failures. We will include this in the future version of the paper.
> 3. The reviewer is correct that for the smaller cohort sizes, full convergence has not occurred. However, the larger cohort sizes have generally all converged. The fairness issues we wish to emphasize are the ones that mirror the generalization issues: Not only do we generalize worse on CIFAR-100/Shakespeare with larger cohorts, we also see that all percentiles of accuracy are reduced, and that are not subject to the convergence issues. We will make this more clear.
> 4. In Algorithm 1 (which we use throughout), clients perform E epochs of training (just as in the original FedAvg paper), and use a fixed batch size. Thus, clients with more data take more training steps. With larger cohorts, we are more likely to sample clients with larger datasets, and therefore are more likely to have the round completion time dictated by the slower clients (ie. the ones with more data) due to the synchronous nature of Algorithm 1. We simulate this slowdown in Appendix B.4. These results mirror previous work on stragglers in distributed learning (eg. Lee et al., “Speeding up distributed machine learning using codes.”).
> 5. The server update norm is not simply a function of how much convergence has occurred. For example, in Figure 6b we see that the server update is roughly constant after round 200. By contrast, FedAvg is far from converging at round 200. In Figure 11 (Appendix B.1) we see that a huge amount of progress is made by FedAvg during these rounds. Additionally, the norm decrease due to increasing the cohort size is one-sided; if this were simply a result of variance, we would likely see at least one instance where the server update norm for a smaller cohort was smaller than the norm for M = 800. This never occurred in our experiments, across many optimizers (save for FedSGD) and many random trials. The behavior is much more consistent with the observation in Figure 6d, where we note that FedAvg is taking an average of nearly orthogonal vectors. The result is (with high probability) very close to 0.
> 6. In our paper, we made two different statements about linear and square root scaling rules. As we state in L265, linear scaling is too aggressive. We do not claim that square root scaling is too aggressive, but we do find that it can improve or worsen performance depending on the task. That being said, tuning the learning rates with the cohort size and algorithm is generally infeasible at the scale of experimentation we used. Moreover, it does not accurately reflect practical FL (which we attempt to capture) where hyperparameter tuning is expensive. Finally, we note that in order to address this issue, we attempted to use a normalized version of FedAvg (Appendix B.9) that helped account for the server update norm issues in Figure 6. We found that it generally had better performance than learning rate scaling, and requires no additional tuning.
> 7. We agree that in some federated settings, we may have to contend with a dynamic population. This introduces relatively unexplored questions of how client populations vary over time though, and to the best of our knowledge this can be extremely application and system-dependent. However, in many cross-device settings, more clients may be available than are needed in a given round. Practical systems often enforce a cohort size (see Bonawitz et al., 2019), so that only M clients are sampled, not all clients who are available. This is to help mitigate total client resource usage. In fact, as we show in Figure 9, our dynamic strategy does help mitigate total client resource usage, especially for lower accuracy thresholds.

---

> > ### Comment · Reviewer_SF2P · 2021-08-20
> > **Thank you for the detailed response**
> >
> > I thank the authors for the detailed response. Many of my questions have been clarified. But I would like to follow up on two points
> >
> > - Section 3.4: I still don't see how your fairness concerns are justified. Fairness is a measure of relative performance across subgroups and if accuracy for all percentiles is reduced why is this concerning in terms of fairness?
> > - I am still not able to follow the reasoning in Section 4 as of why this near orthogonality of the updates should explain the convergence phenomena you see. The near orthogonality of the updates in the later stage of the algorithm seems natural given the way you generate the heterogeneous datasets and the fact that the objectives of the clients are not aligned. You will converge to a solution that is good in average, but every individual client would prefer to improve the model in it's own direction, as long as these updates have a common direction you could make further progress but if they sum to zero there is no room for improvement without personalization. In this case smaller cohort size means fewer terms are active in the sum and thus you get larger norm. But this does not mean more progress on the overall objective, it rather means larger fluctuations.
> > On the other hand, the fact that smaller norms lead to slower convergence leaves some concerns that the step size might just but suboptimal for this setting (which is an orthogonal concern to the effects of the cohort size you aim to study). These effects of stepsize choice should be factored out to make conclusions about the effect of the cohort size. That's why I was asking about the case where you tune it individually for every cohort size to demonstrate the residual effects due to cohort size.
> >
> > Maybe you can do another attempt at rephrasing why you think the observation in Section 4 explains the phenomena you see based on my (mis?)understanding outlined above. So I can be more convinced by this and it might be helpful for other readers as well. Thanks!

---

> > > ### Author Response · Authors · 2021-08-20
> > > **Some clarifications**
> > >
> > > Thank you for the response! Below are some comments:
> > >
> > > **Fairness:** The definition of fairness is nebulous. One definition is relative performance across subgroups (which does not change with cohort size), another is simply the minimum accuracy across subgroups (which can change with cohort size). What we showcase in Section 3.4 is that this minimum accuracy across subgroups is also affected by the generalization issues stemming from the use larger cohorts, but we believe that the observation that relative performance does not change is still an interesting one. We are more than happy to clarify this two-pronged observation: That some notions of fairness get worse, others stay the same.
> > >
> > > **Near-Orthogonality of Client Updates & Convergence:** One important note here is the difference between non-alignment and near-orthogonality; Client updates can be non-aligned but (a priori) one may believe that they still have some alignment, especially given the fact that in many of these datasets (such as EMNIST), all clients have all the labels present in their datasets. The fact that we observe near-orthogonality across all models and datasets is still something that we do not believe is at all obvious.  Also, we stress that the near-orthogonality of client updates primarily explains the "inverse square-root decay" of the server pseudo-gradient. We do not wish to imply (and are happy to clarify in the paper) that it completely determines the convergence, merely that it presents an obstacle to fully utilizing larger cohorts with averaging-based methods.
> > >
> > > **Cohort Size & Convergence:** As for the convergence with smaller cohorts, as we note in the Appendix, we actually tune the learning rates only for the small cohort sizes (specifically, $M = 50$, but we found that tuning for $M = 10$ lead to the same learning rates). Thus, the convergence issues are not due to step-size tuning not being done. If anything, we are disadvantaging the larger cohorts, since we simply borrowed the same learning rate tuned for  $M = 50$.
> > >
> > > Let us know if you have any other questions or concerns. We are more than happy to discuss, and have definitely gained a lot from this conversation.

---

> > > > ### Comment · Reviewer_SF2P · 2021-08-24
> > > > **Thank you for the clarifications**
> > > >
> > > > Regarding the client updates, I think I did not sufficiently appreciate the orthogonality aspect. I was considering near-orthogonality as natural given the heterogeneous data partitioning and the high-dimensional context you are operating in. But I agree, having a sum that vanishes does not per se give you pair-wise orthogonality of the individual terms. And the inverse square-root decay only follows after you realize this.
> > > >
> > > > Regarding the fairness part, I stay with my opinion that calling a small decrease in accuracy across all subgroups a fairness concern is a bit of a stretch. But yeah, it is not wrong if you are specific about the particular fairness notion you are referring to.
> > > >
> > > > I thank the authors for engaging with my concerns. I hope they will take my comments as input for clarifying some statements and explanations in the paper. Overall I am on board accepting the paper given the clarifications by the authors.

---

### Official Review · Reviewer_5Vr7 · 2021-07-16

**Rating:** 7
**Confidence:** 3

**Summary:**

This work introduces a wide empirical study of cohort sizes in local methods for federated learning problems. The authors explain the advantages and disadvantages of using a large cohort number. They provide connections with batch size in centralized optimization. They highlight 5 main aspects: Catastrophic Training Failures, Diminishing Returns, Generalization Failures, Fairness Concerns, and Decreased Data Efficiency. Also, the authors provide several solutions how to overcome issues led by a large cohort number. Authors provide several sequences of experiments with different models, datasets, and methods.

**Limitations And Societal Impact:**

Authors can provide theoretical analysis for a subset of approaches that they provided in section 5.

**Main Review:**

This paper raises many questions regarding using large cohort numbers in federated learning problems. They provide many insights that can be applied to many aspects of machine learning in general. These results are novel and relevant to the Federated Learning community.

Firstly, the authors provide a detailed explanation and good motivation of considered problems. Problems, datasets, and methods are described well.

All experiments have error bars, authors use 5 trials. However, using a large number of trials can be useful.

In section 5 authors propose different ideas to overcome issues that can appear because of the large cohort number. Unfortunately, they do not provide any theoretical analysis. Firstly, the authors propose learning rate scaling. For a similar reason, we can use the clipping technique (https://arxiv.org/pdf/2005.10785.pdf).

Overall, despite the lack of theoretical analysis, this empirical research has good quality. The authors cover many aspects of using large cohort numbers in federated learning. I believe that this paper is good enough to be published.

**Time Spent Reviewing:**

7

---

> ### Author Response · Authors · 2021-08-10
> **Response to Reviewer 5Vr7**
>
> Thank you for your time and positive assessment of our work. We agree that a larger number of trials would be beneficial. However, given the breadth and scope of our experiments, performing significantly more trials than 5 would be computationally infeasible. Moreover, we found that in many experiments, the standard deviation (beyond a cohort size of 1) was close to zero.
>
> We also agree with the reviewer that we would like to see a theoretical analysis of some of the results in our paper. However, we wish to emphasize that understanding non-federated large batch training methods theoretically is still challenging, despite the years of work in this area. We hope that our paper can mirror that of works such as (Keskar et al., 2017) and serve as a rigorous empirical foundation on which to build theory. In particular, we attempted to empirically document and explore open questions that are ripe for theoretical analysis, such as the near orthogonality of client updates. Given the large gap between theory and practice in federated learning (as exemplified by the fact that many analyses of FedAvg do not attain better convergence rates than mini-batch SGD), we believe that a theoretical analysis is beyond the scope of this work, and would be better addressed in work that can more narrowly focus on a single facet of large-cohort training.

---

> > ### Comment · Reviewer_5Vr7 · 2021-08-19
> > **My reply to authors**
> >
> > Thank you for your clarifications!

---

> > ### Comment · Reviewer_5Vr7 · 2021-08-25
> > **Dynamic cohort sizes**
> >
> > I have a question about strategy of Dynamic cohort sizes. In your paper you "start with an initial cohort size of M = 50 and double the size every 300 rounds up to M = 800 (or the maximum population size if smaller)."
> >
> > Can you please explain why you used this strategy? Did you try any other approaches? Is it possible to find a good strategy for the problem in advance (without Brute-force search)?

---

> > > ### Author Response · Authors · 2021-08-25
> > > **On Dynamic Cohort-Size Strategies**
> > >
> > > Thanks for the question, this is definitely important. Our primary focus in Section 5 was to determine whether methods for improving large-batch centralized training could be immediately adapted to cohort sizes in federated learning. For the dynamic cohort-sizes, we focused on what is (to the best of our knowledge) one of the most empirically effective dynamic batch size strategies for large-batch centralized training: The "doubling mechanism" from Smith et al., "Don't Decay the Learning Rate, Increase the Batch Size." This method doubles the batch size every $E$ epochs, for some value of $E$. Thus, it was natural to consider a federated version where we double the cohort size every $R$ rounds, for some value of $R$.
> > >
> > > As for other approaches, we found that this doubling strategy performed better than just about all other methods we tried. Note that this is true in centralized learning: To the best of our knowledge, the doubling mechanism from Smith et al., 2017 performs near optimally among most methods that increase batch size over time. We did not brute-force search any kind of mechanism of this ilk, rather we relied on the literature on large-batch training to inform our methods, as we are interested in whether such methods carry over to federated learning.
> > >
> > > We wish to stress that this strategy is probably not completely optimal for federated learning. However, trying to optimize it could take the span of an entirely new work (as evidenced by the fact that Smith et al., 2017 devote an entire paper to this question in centralized settings).

---

> > > > ### Comment · Reviewer_5Vr7 · 2021-09-02
> > > > **On Dynamic Cohort-Size Strategies**
> > > >
> > > > Thank you for you clarifications! I think this work provides a wide range of experimental results. These results are insightful and informative. I understand that it is impossible to cover all aspects in one paper, so I hope this work can be influential and topics introduced in this paper will be explored further. Although there is no any theory, I believe this work should be published.

---

### Official Review · Reviewer_Lnmy · 2021-07-17

**Rating:** 7
**Confidence:** 5

**Summary:**

This paper investigates the impact of the size of the sampled client (called cohort size) on the quality of the learned model and the training dynamics in FL via extensive empirical study. It demonstrates several particular issues are were not observed in the centralized large-batch training. It also provides an empirical explanation and some heuristic approaches to solve the issues based on the characteristics of federated training dynamics.



**Limitations And Societal Impact:**

yes

**Main Review:**

The problem of the cohort size in FL is well-motivated, as similar issues have also occurred in classical large-batch training. The experiments are solid. The near-orthogonality observations are interesting and shed light on the impact of adaptive optimizers.

I hope to see more future work that solves the problem of large batch size training (still open even for centralized cases).

I suggest the authors consider using the static Batch Normalization technique proposed in the following reference, which significantly outperforms other normalization methods (including the currently used GroupNorm) in FL settings.
"HeteroFL: Computation and Communication Efficient Federated Learning for Heterogeneous Clients" by Diao et al.



**Time Spent Reviewing:**

1 hour

---

> ### Author Response · Authors · 2021-08-10
> **Response to Reviewer Lnmy**
>
> Thank you for the positive assessment of our work and for the suggestions on improvements. We agree that even understanding large batch training in centralized settings is generally still an open problem, and we hope that our work can help open the door to understanding large cohort federated training. In particular, we believe that our work provides a federated mirror to much of the empirical work (such as Keskar et al., 2017) that set the stage for understanding large-batch centralized training.
>
> We agree with the reviewer that the static Batch Normalization technique by Diao et al. seems extremely promising in federated learning. While we believe that work attempting to achieve state-of-the-art results on federated tasks should consider using this technique, we wish to emphasize that our goal was not to advance state-of-the-art results on any given task. Rather, we selected models and datasets that were well represented in the existing FL literature, in order to showcase just how much the cohort size impacts previous work in this area. We used GroupNorm in our ResNet model due to its prevalence in the literature (eg. see Hsieh et al., 2019 and Reddi et al., 2021, among others), as this better allowed us to isolate the impacts of changing the cohort size alone. That being said, we hope to see the impact of cohort size on static Batch Normalization layers (and other model architectures designed for FL) in the future.

---

> > ### Comment · Reviewer_Lnmy · 2021-08-19
> > **Response to authors**
> >
> > The authors have addressed my comments. I suggest the authors incorporate the related discussions in their revision.

---

### Decision · Program_Chairs · 2021-09-27

**Decision:**

Accept (Poster)

**Comment:**

The work is a well written and well executed empirical study of phenomena associated with large cohort sizes when training practical cross-device FL models.

All reviewers ultimately recommended to accept this paper for publication, and I concur. Moreover, I enjoyed the rich discussion between the reviewers and the authors. I have read the paper in detail myself, and have enjoyed the experience. However, since no theoretical explanation is offered, and because the scope of the experiments is necessarily limited, I am not fully convinced about the robustness of the conclusions. Because of this, I consider this paper to be a borderline accept.

---

A few caveats:

- I believe that some of these observations may have natural theoretical explanations from existing/known theory (granted, under some constraining assumptions on the models and losses trained), which the authors are not exploring.

- Given the experimental setup, whatever the results would be, *something* would certainly be observed, and a commentary on that something could always be made. Would that mean that a pattern or a phenomenon was discovered? Maybe, but not necessarily. Many more experiments across a vast array of model types and sizes would be needed for a more convincing argument to be made. It is very hard to say whether the presented observations are robust enough - would they be observed for other models and datasets? What would happen, for example, if you experimented with simple linear models that lead to convex optimization problems? What happens if local steps are removed and one uses compressed communication instead?

Having said that, the observations, however fragile, are useful for further theoretical and empirical studies, and when replicated by other researchers in other settings, may serve as an inspiration to design practical mitigation strategies, such as those outlined in the work.